# Exact Expressive Power of Transformers with Padding

**William Merrill**[*]
Allen Institute for AI
willm@allenai.org

**Ashish Sabharwal**
Allen Institute for AI
ashishs@allenai.org

## Abstract

Chain of thought is a natural inference-time method for increasing the computational power of transformer-based large language models (LLMs), but comes at the cost of sequential decoding. Are there more efficient alternatives to expand a transformer's expressive power without adding parameters? We consider transformers with *padding* tokens as a form of parallelizable test-time compute. We show that averaging-hard-attention, masked-pre-norm transformers with polynomial padding recognize precisely the class FO-uniform $\mathsf{TC}^0$ of extremely parallelizable problems. While the $\mathsf{TC}^0$ upper bound was known, proving a matching lower bound had been elusive. Further, our novel analysis reveals the precise expanded power of padded transformers when coupled with another form of inference-time compute, namely dynamically increasing depth via *looping*. Our core technical contribution is to show how padding helps bring the notions of *complete problems* and *reductions*, which have been a cornerstone of classical complexity theory, to the formal study of transformers. Armed with this new tool, we prove that padded transformers with $\mathrm{O}(\log^d n)$ looping on inputs of length $n$ recognize exactly the class FO-uniform $\mathsf{TC}^d$ of moderately parallelizable problems. Thus, padding and looping together systematically expand transformers' expressive power: with poly-logarithmic looping, polynomially padded transformers recognize precisely the class FO-uniform $\mathsf{NC}$, the best that could be expected without losing parallelism (unless $\mathsf{NC} = \mathsf{P}$). Our results thus motivate further exploration of padding and looping as parallelizable alternatives to chain of thought for test-time compute.

## 1 Introduction

Due to the computational limitations of transformers (Merrill and Sabharwal, 2023a; Strobl et al., 2024; Chiang, 2025), solving complex reasoning problems requires extending their computational power at inference time, typically by allowing models to generate long chains of thought (CoT) before their outputs (Wei et al., 2022; Nye et al., 2022). Theoretical work has shown how CoT expands the expressive power of transformers to sequential problems outside the class $\mathsf{TC}^0$ of highly parallelizable problems, but it also sacrifices parallelism (Merrill and Sabharwal, 2023b; Li et al., 2024), making inference slow. Are there alternative inference-time compute approaches that increase expressive power while preserving parallelism?

One method for parallelizable inference-time compute with LLMs is using *padding tokens* rather than CoT. Padding can be understood as restricted CoT where the tokens on the chain are restricted to some "blank" symbol rather than tokens generated by the LLM. Since all the input tokens are known in advance, padding is more parallelizable than CoT. There have been some attempts to make padding practical with mixed results (Goyal et al., 2024; Pfau et al., 2024), but it is not fully understood. Specifically, while it is known that padded transformers remain in $\mathsf{TC}^0$, it has been open and elusive whether they can solve *all* problems in $\mathsf{TC}^0$ or even the smaller class $\mathsf{AC}^0$ (Pfau et al., 2024).

---

[*]Work partially conducted as a PhD student at New York University.

39th Conference on Neural Information Processing Systems (NeurIPS 2025).

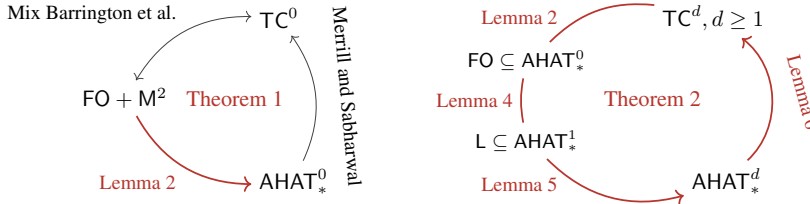

Figure 1: Summary of core results: exact characterizations of the expressive power of $O(\log^d n)$-depth looped AHATs with padding, for $d \geq 0$. Theorem 1 shows that $\mathsf{AHAT}^0_* = \mathsf{FO}$-uniform $\mathsf{TC}^0$. Theorem 2 extends this to show that, for $d \geq 0$, $\mathsf{AHAT}^d_* = \mathsf{FO}$-uniform $\mathsf{TC}^d$. In the process of obtaining these results, we also found the novel circuit complexity result that, for any $d \geq 1$, $\mathsf{FO}$-uniform $\mathsf{TC}^d = \mathsf{L}$-uniform $\mathsf{TC}^d$ Theorem 3. Thus, for $d \geq 1$, $\mathsf{AHAT}^d_* = \mathsf{L}$-uniform $\mathsf{TC}^d$.

Our first contribution is exactly characterizing the expressive power of transformers[2] with polynomial padding as $\mathsf{FO}$-uniform $\mathsf{TC}^0$, answering an open question (Pfau et al., 2024). The result emerges through a finer-grained analysis of transformers in terms of string logics (Merrill and Sabharwal, 2023c; Chiang et al., 2023; Yang et al., 2024; Yang and Chiang, 2024). Thinking in terms of logic, we show that $n^k$ padding tokens give transformers enough "storage space" to resolve any first-order majority logic formula over $k$ variables. This suffices to capture all of $\mathsf{FO}$-uniform $\mathsf{TC}^0$ (Mix Barrington et al., 1990), giving an exact characterization of the expressive power of such transformers.

This first result, however, does not clarify whether padded transformers can be minimally extended to gain expressivity *beyond* $\mathsf{TC}^0$. To address this, we consider the combination of padding with *looping*, i.e., repeating a block of layers dynamically as a function of input length, also referred to as *universal* transformers (Dehghani et al., 2019; Giannou et al., 2023). This can be understood as a form of inference-time compute where padding controls the computation width and looping controls the computation depth—crucially, without adding any parameters. If the number of repetitions is minimal (e.g., sublinear) in sequence length, this transformer model remains highly parallelizable relative to CoT (Merrill and Sabharwal, 2025).

Extending this result, we show that, for $d \geq 1$, transformers with polynomial padding and $\mathrm{O}(\log^d n)$ looping recognize exactly $\mathsf{FO}$-uniform $\mathsf{TC}^d$. Thus log or polylog-looped transformers can solve many problems outside $\mathsf{TC}^0$ under standard complexity conjectures, including boolean formula evaluation, iterated matrix multiplication, graph connectivity, and context-free language recognition. It also follows that polylog-looped padded transformers converge in expressive power to $\mathsf{FO}$-uniform $\mathsf{NC}$, the ceiling that could be expected while preserving parallelism under standard complexity conjectures.

Our first result about the expressive power of fixed-depth padded transformers provides crucial insight for our extended results about looped padded transformers (cf. Figure 1), by allowing, for the first time, the use of familiar tools of *reductions* and *complete problems* from classical complexity theory in the analysis of transformers. First, the ability to express $\mathsf{TC}^0$ implies that padded transformers can implement $\mathsf{FO}$ reductions. We then scaffold the ability to implement $\mathsf{FO}$ reductions to show that transformers can also implement $\mathsf{L}$ reductions, via known results about transformers' ability to implement graph connectivity, an $\mathsf{NL}$-complete problem under $\mathsf{FO}$ reductions. We then use this to show that $\mathrm{O}(\log^d n)$ looping recognizes exactly $\mathsf{FO}$-uniform $\mathsf{TC}^d$, via the "wide" $\mathsf{TC}^d$ circuit evaluation problem, which we prove to be complete for this class under $\mathsf{L}$ reductions.

Finally, we prove a novel circuit complexity result that $\mathsf{FO}$-uniform $\mathsf{TC}^d = \mathsf{L}$-uniform $\mathsf{TC}^d$ for any $d \geq 1$. This was obtained in our analysis of looped transformers with $d \geq 1$. Besides being potentially of independent interest in circuit complexity, this allows us to see that, for $d \geq 1$, $\mathrm{O}(\log^d n)$ looping allows recognizing not just $\mathsf{FO}$-uniform $\mathsf{TC}^d$ but also $\mathsf{L}$-uniform $\mathsf{TC}^d$ (because they are the same).

Overall, our results provide an *exact characterization* of the expressive power of padded fixed-depth transformers as $\mathsf{TC}^0$, answering an open question (Pfau et al., 2024). Further, we exactly

---

[2]Our formal model, detailed in Section 2, assumes a fully uniform transformer with fixed parameters, fixed width, logarithmic or polynomial precision (showing their equivalence in this setting), masked pre-norm, and causal masking (Theorem 1 also applies to unmasked transformers, showing their equivalence under padding).

characterize the expressive power of polylog-looped padded transformers, which reveals that padding and looping dramatically extend the expressive power of transformers under standard complexity conjectures. We take these results to motivate empirical investigation of padding and looping as forms of inference-time compute that are more parallelizable than standard CoT.

## 2 Preliminaries

### 2.1 Averaging-Hard-Attention Transformers

Following previous work, we analyze a model of transformers that is slightly idealized compared to standard soft-attention transformers. Specifically, we analyze transformers with averaging hard attention (AHAT; also called "saturated" by Merrill et al., 2022) and masked pre-norm (Merrill and Sabharwal, 2023b). AHAT attention heads only attend to the values that *maximize* the attention score; in the case of ties, the head returns a uniform average of all values with tied scores. Masked pre-norm means that the transformer can apply a linear projection before pre-norm at the beginning of each sublayer, which is useful for reading individual values from the residual stream without interference from other values. Overall, we believe these idealizations are minimal changes that make it easier to implement algorithmic constructions that generalize to any sequence length.

More formally, we define the transformer sublayers in this AHAT model following Merrill and Sabharwal (2025). Each sublayer uses masked pre-norm (Xiong et al., 2020; Merrill and Sabharwal, 2023b), reading input $\mathbf{z}_i = \mathsf{layer\_norm}(\mathbf{M}\mathbf{h}_i)$, where $\mathbf{h}_i$ is the previous sublayer output, $\mathbf{M}$ is some matrix, and layer-norm can be standard layer-norm (Ba et al., 2016) or RMS norm (Zhang and Sennrich, 2019). The sublayer outputs $\delta_1, \ldots, \delta_n$, and the residual stream is updated as $\mathbf{h}'_i = \mathbf{h}_i + \delta_i$.

**Definition 1** (Self-attention sublayer). The self-attention sublayer is parameterized by a mask $\mathbf{m} \in \mathbb{Q}^m$, output projection matrix $\mathbf{W} : \mathbb{Q}^m \to \mathbb{Q}^m$, and, for $1 \leq k \leq h$, query, key, and value matrices $\mathbf{Q}^k, \mathbf{K}^k, \mathbf{V}^k$, each of which is a projection from $\mathbb{Q}^m$ to $\mathbb{Q}^{m/h}$.

Given input $\mathbf{z}_i$, the self-attention sublayer computes queries $\mathbf{q}_i = \mathbf{Q}^k \mathbf{z}_i$, keys $\mathbf{k}_i = \mathbf{K}^k \mathbf{z}_i$, and values $\mathbf{v}_i = \mathbf{V}^k \mathbf{z}_i$. Next, these values are used to compute the attention head outputs:

$$\mathbf{a}_{i,k} = \lim_{\tau \to 0} \sum_{j=1}^{c} \frac{\exp(1/\tau \cdot \mathbf{q}_{i,k}^{\top} \mathbf{k}_{j,k})}{Z_{i,k}} \cdot \mathbf{v}_{j,k}, \quad \text{where } Z_{i,k} = \sum_{j=1}^{c} \exp\left(1/\tau \cdot \mathbf{q}_{i,k}^{\top} \mathbf{k}_{j,k}\right).$$

We can set $c = i$ to define *causally masked* attention and $c = n$ for *unmasked* attention. Averaging hard attention is formalized by taking the low-temperature limit ($\tau \to 0$), which causes all probability mass to be concentrated on the tokens that maximize the attention score. In practice, transformers can approximate this by learning a temperature close to zero; for a fixed sequence length, this approximation will hold, but for longer strings it will break down. Finally, the output of the self-attention sublayer is computed by aggregating the head outputs via $\delta_i = \mathbf{W} \cdot \mathrm{concat}(\mathbf{a}_{i,1}, \ldots, \mathbf{a}_{i,h})$.

**Definition 2** (Feedforward sublayer). The feedforward sublayer at layer $\ell$ is parameterized by a mask $\mathbf{m} \in \mathbb{Q}^m$ and projections $\mathbf{W} : \mathbb{Q}^m \to \mathbb{Q}^w$ and $\mathbf{U} : \mathbb{Q}^w \to \mathbb{Q}^m$.

A feedforward layer computes a local update to the residual stream via $\delta_i = \mathbf{U} \cdot \mathsf{ReLU}(\mathbf{W}\mathbf{z}_i)$.

A transformer defines a function $\Sigma^* \to \Sigma$ or $\Sigma^* \to \{0, 1\}$ if we add a linear head to the final layer and take its argmax as the output. We say that a transformer $T : \Sigma \to \{0, 1\}$ recognizes a language $L$ (with beginning-of-sequence token \$) if, for any $w \in \Sigma^*$, $T(\$w) = 1$ if and only if $w \in L$.

**Precision.** In Appendix A, we formalize logarithmic (Merrill and Sabharwal, 2023a) and polynomial precision (Chiang, 2025) datatypes. All our constructions go through with either datatype, showing that logarithmic and polynomial-precision looped padded transformer classes are, in fact, identical.

**Layer-Norm Hash.** Our constructions will use the layer-norm hash representation (Merrill and Sabharwal, 2023b, 2025) for query-key matching with positive integer values.

**Definition 3.** Given $z \in \mathbb{N}$, its *layer-norm hash* is the vector $\langle z, 1, -z, -1 \rangle / \sqrt{2z^2 + 2}$.

The layer-norm hash is computable within a transformer and satisfies that property that $\phi(i)^{\top} \cdot \phi(j) = 1$ if and only if $i = j$. This makes it useful for retrieval based on position matching using AHATs.

## 2.2 Padded and Looped AHATs

We assume a looped transformer model as previously defined by Merrill and Sabharwal (2025):

**Definition 4** ($d(n)$-looped transformer). A looped transformer's layers are partitioned into blocks $\langle A, B, C \rangle$. On an input of length $n$, block $B$ is repeated depth-wise $d(n)$ times.

Looped transformers provide a way of dynamically scaling width at test time by looping a block of layers. In addition, we also consider *padding* (Pfau et al., 2024) as a simple way to increase width:

**Definition 5** ($w(n)$-padded transformer). On an input of length $n$, we first append $w(n)$ "blank" tokens ($\square \notin \Sigma$) and then feed this full string through the transformer.

Looping and padding can also be combined, giving us the following language class for transformers:

**Definition 6** (Padded and looped transformers). Let $d, k \in \mathbb{Z}_{\geq 0}$. $\mathsf{AHAT}_k^d$ is the class of languages recognizable by a causally masked looped transformer with masked pre-norm, a beginning-of-sequence token, no position embedding, $\mathrm{O}(\log^d n)$ depth, and $\mathrm{O}(n^k)$ padding tokens. Further, $\mathsf{AHAT}_*^d = \bigcup_{k=0}^{\infty} \mathsf{AHAT}_k^d$, $\mathsf{AHAT}_k^* = \bigcup_{d=0}^{\infty} \mathsf{AHAT}_k^d$, and $\mathsf{AHAT}_*^* = \bigcup_{d=0}^{\infty} \mathsf{AHAT}_*^d$.

$\mathsf{AHAT}_k^d$ uses causal masking with no positional encodings (but with a beginning-of-sequence (BoS) token \$; cf. Merrill and Sabharwal, 2023b). In contrast, some of our constructions will be for **unmasked transformers**, which must use position encodings to distinguish positions. We will write $\mathsf{uAHAT}_k^d$ for the language classes recognizable by unmasked transformers with $1/i$ position encodings. We will occasionally consider **mixed-masked transformers** where some attention heads use causal masking and others do not. These transformers, which do not need position encodings since they can compute $1/i$ with causal heads and the BoS token, will be denoted $\mathsf{mAHAT}_k^d$.

With some abuse of notation, we will use $\mathsf{AHAT}_k^d$ to denote both the class of languages as defined above as well as the corresponding class of transformer models, i.e., transformers with $\mathrm{O}(\log^d n)$ and $(n^k)$ padding tokens. The distinction should be clear from the context.

## 2.3 Circuit Complexity

We define the circuit complexity classes $\mathsf{AC}^d$ and $\mathsf{TC}^d$ in the standard way (Arora and Barak, 2009; Strobl et al., 2024). A circuit family is a mapping $\mathcal{C} = \{C_n\}_{n=0}^{\infty}$ from input lengths $n$ to circuits that take $n$ inputs. $\mathsf{AC}^d$ is the class of language that can be recognized by polynomial-size, fixed-depth circuit families with unbounded-arity AND/OR gates. $\mathsf{TC}^d$ is the same class but augmented with unbounded-arity MAJ gates that take a majority vote over their input bits.

It will often be useful to talk about *uniform* variants of these classes. A X-uniform circuit family obeys that constraint that circuit $C_n$ for input size $n$ can be "built" from the input string $1^n$ using computation in X. We will consider two standard notions of uniformity: FO uniformity (which is equivalent to DLOGTIME uniformity; Mix Barrington et al., 1990) and the weaker L uniformity (i.e., log-space uniformity). For a circuit class XC, we will write A-uniform XC to denote that class constrained to circuit families satisfying A uniformity. See Appendix B for formal definitions of each type of uniformity and Strobl et al. (2024) for further context.

Finally, we will also use the notion of completeness for circuit classes. Informally, an X-complete problem is a problem in X to which any other problem in X can be mapped via some simple reduction R. Whether a problem is complete depends on the notion of reduction used. See Definition 9 for a more formal definition of reductions, which makes completeness fully defined.

## 2.4 Logic

We define the standard notion first-order logic over strings (FO; cf. Merrill and Sabharwal, 2023c). FO formulas map strings to boolean values. Its formulas can check for token occurrences at specific string positions, and it allows quantification over positions in the input string. More formally:

**Definition 7.** FO contains two types: indices, representing positions in the input string, and formulas, which evaluate to true or false. For an index or formula $x$, we write $[\![x]\!]^{w,v}$ for the evaluation of $x$ on string $w$ with variable assignments $v$ (a map from names to values). Indices in FO are integers denoting positions in the input string:

1. The constant 1, representing the first token's position: $[\![1]\!]^{w,v} = 1$.
2. The constant $n$, representing the last token's position: $[\![n]\!]^{w,v} = |w|$.
3. Symbols (e.g., $i, j, k$) representing variables ranging over positions 1 to $n$: $[\![i]\!]^{w,v} = v[i]$.

Formulas in FO are then constructed as follows:

1. Let $\Sigma$ be a finite alphabet. For each $\sigma \in \Sigma$ and any index $i$, $Q_\sigma(i)$ is a formula that is true when the $i$-th input token is $\sigma$. That is, $[\![Q_\sigma(i)]\!]^{w,v} = 1$ iff $w_m = \sigma$ where $m = [\![i]\!]^{w,v}$.
2. For two indices $i, j$, $i = j$, $i \le j$, and $i \ge j$ are formulas with their conventional semantics.
3. For two indices $i, j$, $\mathrm{bit}(i, j)$ is a formula returning the $j$-th bit of $i$.
4. For two formulas $P, Q$, $P \wedge Q$ and $P \vee Q$ are formulas with their conventional semantics.
5. For any formula $P$ (which may refer to $i$ or any over variable), the following are formulas:
   (a) $\exists i.P$ means setting $i$ to some value $m \in [1, n]$ makes $P$ true (more formally, $[\![P]\!]^{w,v|i=m} = 1$).
   (b) $\forall i.P$ means setting $i$ to any value $m \in [1, n]$ make $\phi$ true.

An FO formula $P$ with no free variables is called a *sentence* and returns a value in $\{0, 1\}$ for each input string. The language *defined* by a sentence is the set of strings mapped to 1. The *nesting depth* of an FO formula is the depth of its syntactic tree constructed by the rules above. The *number of distinct variables* is the number of different variable names used in $P$. This can be minimized by allowing two independent quantifiers in parallel subformulas to use the same variable name.

It is known that the class of languages defined by FO is exactly the circuit complexity class DLOGTIME-uniform $\mathsf{AC}^0$ = FO-uniform $\mathsf{AC}^0$ (Mix Barrington et al., 1990). This class captures the languages recognized by unique hard-attention transformers (Hao et al., 2022; Yang et al., 2024), but it is not large enough to capture soft-attention transformers (Merrill and Sabharwal, 2023a). To consider a more expressive logic capable of modeling soft-attention transformers, we can extend FO with majority quantifiers (Merrill and Sabharwal, 2023c). Specifically, we will define $\mathsf{FO + M}^2$ as FO extended to include a *paired majority* quantifier:

**Definition 8** (Mix Barrington et al., 1990). $\mathsf{FO + M}^2$ is FO extended to include the $\mathsf{M}^2$ quantifier, defined as follows: $\mathsf{M}^2(i, j).P(i, j)$ is true if $\phi(i, j)$ holds for a majority of *pairs* of positions $(i, j) \in [n]^2$ in the string.

It is known that $\mathsf{FO + M}^2$ defines exactly FO-uniform $\mathsf{TC}^0$ (Mix Barrington et al., 1990, Theorem 10.2, which uses paired majority). It is also possible to use majority quantifiers to define addition over indices, so without loss of generality, we can assume $\mathsf{FO + M}^2$ formulas have no addition (unlike FO formulas). Moreover, it is possible to simulate bit in terms of $\mathsf{M}^2$, so we can consider $\mathsf{FO + M}^2$ formulas not to contain bit, in contrast to the more standard logic $\mathsf{FO + M[bit]}$ that also defines $\mathsf{TC}^0$. This makes $\mathsf{FO + M}^2$ a simpler target for our transformer constructions than $\mathsf{FO + M[bit]}$.

## 3 Masked and Unmasked Transformers

Before presenting our main results, we briefly discuss a simple but important relationship between different kinds of masking that will come in handy. When using a transformer as a recognizer for formal languages, there is a choice of whether to use an encoder (with no attention masking) or a decoder (with causal attention masking) to encode the input string. Causally masked (decoder) models are more standard these days, as well as theoretically more convenient (e.g., no need for position embedding, as position information can be derived from causal attention), so we take them as our default model. However, it is sometimes easier to reason about how to solve problems with unmasked transformers. Fortunately, the following lemma shows we can simulate an unmasked transformer (encoder) with a causally masked transformer (decoder), if we allow padding tokens proportional to the transformer depth. This will be useful going forward in several places where we convert unmasked and mixed-masked constructions to causally masked constructions.

**Lemma 1** (Unmasked to Causally Masked). *Let $E$ be an unmasked (with position encoding $1/i$) AHAT encoder with depth $\ell \ge 1$. Then there exists a causally masked AHAT decoder $D$ (without any position encoding) with depth $\ell$ and with $\ell n$ padding tokens on input length $n$ that is equivalent to $E$ in the following sense: the final $n$ outputs of $D$ match the original $n$ outputs of $E$.*

*Proof.* We first observe that the causally masked decoder $D$ can compute $1/i$ (the position encoding used in the unmasked encoder $E$) by attending uniformly with value 1 only for the beginning-of-

sequence symbol. To simulate unmasked attention with causally masked attention via padding, the idea is for $D$ to unroll a sequence of $\ell$ "blocks" (one for each of the $\ell$ layers of $E$) along the padding length dimension, each of width $n$. Each block will attend to the previous block in the previous layer and can thus see all tokens despite causal masking.

To implement the block construction, we first compute $\phi(n)$. We then use this to compute $\phi(b_i)$, where $b_i = \lfloor i/n \rfloor \in [1, \ell]$, which represents the block that each token belongs to. We also compute $\phi(b_i - 1) \in [0, \ell - 1]$ using a head that ignores the beginning-of-sequence token \$. Next, we modify each original head from $U$ to have an additional term in the query/key product. The query is $C\phi(b_i - 1)$ and the key is $\phi(b_j)$, where $C$ is a large weight. We set $C$ to a fixed large value (independent of $n$) such that this term dominates all other terms in the inner product computation at each token. As a result, the head is constrained over keys in the context where $b_j = b_i - 1$, i.e., the keys from the last block. Within these keys, it computes exactly the same AHAT output as the original unmasked head. In this way, we are able to simulate an unmasked (or mixed mask) transformer with a causally masked transformer. □

Since causally masked heads of a mixed-masked transformer can be trivially simulated by a causally masked transformer, Lemma 1 allows us to relate various masking variants (proof in Appendix C):

**Proposition 1.** *Unmasked (with position encoding $1/i$ or $i/n$; cf. Lemma 10 in §C) and mixed-masked padded transformers (i.e., uAHATs and mAHATs) can be simulated by causally masked transformers (AHATs). Specifically, for any $d, k \in \mathbb{N}$, the following holds for the corresponding problem classes:*

1. *$\mathsf{uAHAT}_k^0 \subseteq \mathsf{mAHAT}_k^0 \subseteq \mathsf{AHAT}_{\max\{k,1\}}^0$*
2. *$\mathsf{uAHAT}_k^d \subseteq \mathsf{mAHAT}_k^d \subseteq \mathsf{AHAT}_{1+\max\{k,1\}}^d$ for $d \geq 1$*
3. *$\mathsf{uAHAT}_*^d \subseteq \mathsf{mAHAT}_*^d \subseteq \mathsf{AHAT}_*^d$.*

## 4 Fixed-Depth Padded Transformers Recognize Exactly FO-uniform $\mathsf{TC}^0$

We are now ready to prove our first main result, namely that padding tokens allow transformers to simulate FO formulas—with more padding allowing more nesting of variables. Moreover, they can simulate formulas with the special *paired majority* quantifiers.

**Lemma 2.** *An $\mathsf{FO} + \mathsf{M}^2$ formula with $k \geq 1$ distinct variables and nesting depth $\ell$ can be computed in $\mathsf{uAHAT}_k^0$ (and hence in $\mathsf{AHAT}_k^0$) with (fixed) depth $\ell$.*

*Proof.* We will store all $n^k$ possible configurations of $k$ variables of the formula using $n^k$ padding tokens, where each token corresponds to a specific configuration of all the variables, which we denote $v$ (cf. Definition 7). We will present an inductive transformer construction that uses a single layer to compute the boolean variable of each formula and the integer value of each numerical expression in $\mathsf{FO} + \mathsf{M}^2$, assuming the constituent formulas and values were already computed at the previous layer. Since each $\mathsf{FO} + \mathsf{M}^2$ formula consists of a fixed number of subformulas and values, we can accumulate all constituents in the residual stream in order to compute any larger formula.

In more detail, let $[\![x]\!]^{w,v}$ be the value of a formula or numerical value $x$ on string $w$ under assignment $v$, with $k$ total variables. We will identify each variable assignment with an integer $v \in [n^k]$, so that, for every $P$, padding token $v$ stores $[\![P]\!]^{w,v}$ in the residual stream with a scalar whose sign indicates a truth value. For a numerical value $i$, token $v$ will represent $[\![i]\!]^{w,v}$ as a small vector $\phi([\![i]\!]^{w,v})$. where $\phi$ is the layer-norm hash (Definition 3). We show how to evaluate each constituent of an $\mathsf{FO} + \mathsf{M}^2$ formula. As mentioned after Definition 8, addition over indices and bit are subsumed by $\mathsf{M}^2$, so we do not need to simulate them.

1. Constants. To compute the constant $1$ or $n$ at each configuration $v$, we can attend from each $v$ to the first and last input token and retrieve its position to get $\phi(1)$ or $\phi(n)$, where $n = |w|$.

2. Variables. At each token $v$, we will compute $\phi([\![i]\!]^{w,v})$, a representation of the value of variable $i$ under assignment $v$. We view the integer $v \in [n^k]$ as a tuple of $k$ values $v[1], \ldots, v[k] \in [n]$. We have that $[\![i]\!]^{w,v} = v[i]$. To get this, we first compute $\phi(v)$ and then take a "projection" to retrieve $\phi(v[i])$ using quotient and remainder operations (Merrill and Sabharwal, 2025, Lemma 3.1). More

formally, we compute $\phi(n)$ where $n = |w|$, divide $i - 1$ times by $n$, and then take the remainder $\mod n$ to obtain:

$$\phi(\lfloor v/n^{i-1} \rfloor \mod n) = \phi(v[i]).$$

Thus, we conclude that we can compute $\phi(v[i]) = \phi(\llbracket i \rrbracket^{w,v})$ at each token $v$.

3. Comparisons. Given two numerical values $i, j$ (either variables or constants) already stored at $v$ as $\phi(\llbracket i \rrbracket^{w,v)}, \phi(\llbracket j \rrbracket^{w,v})$, we can simply compare $\phi(\llbracket i \rrbracket^{w,v}) - \phi(\llbracket j \rrbracket^{w,v})$ on some axis and apply layer-norm to obtain $\llbracket i > j \rrbracket^{w,v}$ at token $v$.

4. Token Predicates. Assume we have $\phi(\llbracket i \rrbracket^{w,v})$ previously stored at assignment $v$. We can hard-attend to retrieve token $w_m$ where $m = \llbracket i \rrbracket^{w,v}$. We then compute $\llbracket Q_\sigma(i) \rrbracket^{w,v}$ by checking whether $w_m = \sigma$.

5. Connectives. Assume we have $\llbracket P \rrbracket^{w,v}$ and $\llbracket Q \rrbracket^{w,v}$ previously stored at each assignment token $v$. Then we can simply use a feedforward network to compute $\llbracket \neg P \rrbracket^{w,v}$, $\llbracket P \wedge Q \rrbracket^{w,v}$, or $\llbracket P \vee Q \rrbracket^{w,v}$ independently at each $v$.

6. Standard Quantifiers. Assume we have $\llbracket P \rrbracket^{w,v}$ stored at each configuration $v$. Then, at each $m$, we want to resolve the quantifier $Q$ over the set $\{\llbracket P \rrbracket^{w,v|i=m}\}_{m=1}^n$, where $v|i = m$ denotes $v$ with $i$ overriden to have value $m$. We will count $c$, the number of $m$ such that $P^{v|i=m}$ holds. More formally, let $j_1, \ldots, j_{k-1}$ be the set of variables excluding $i$. Then we can attend from $v$ over $v'$ with query $\langle \phi(j_1^v), \ldots, \phi(j_{k-1}^v) \rangle$, key $\langle \phi(j_1^{v'}), \ldots, \phi(j_{k-1}^{v'}) \rangle$, and value $P^{v'}$ to retrieve $c/n$. Finally, we threshold $c/n$ against $\frac{1}{2n}$ (for $\exists$) or $\frac{2n-1}{2n}$ (for $\forall$) to resolve $\llbracket Q i.\phi \rrbracket^{w,v}$.

7. Paired Majority Quantifiers. Assume we have $\llbracket P \rrbracket^{w,v}$ stored at each configuration $v$. We want to compute $\llbracket M(i, j).P \rrbracket^{w,v}$ for any two variables $i, j$ already represented. The idea slightly generalizes the construction for single-variable quantifiers: we will use attention to count $c$, the number of assignments where $\llbracket P \rrbracket^{w,v|i=m,j=\ell}$ is true. Formally, let $j_1, \ldots, j_{k-2}$ be the set of variables excluding $i, j$. Then we can attend from $v$ over $v'$ with query $\langle \phi(j_1^v), \ldots, \phi(j_{k-2}^v) \rangle$, key $\langle \phi(j_1^{v'}), \ldots, \phi(j_{k-2}^{v'}) \rangle$, and value $\llbracket P \rrbracket^{w,v'}$ to retrieve $c/n^2$. Finally, we threshold $c/n^2$ against $\frac{1}{2}$ to resolve $\llbracket M^2(i, j).P \rrbracket^{w,v}$.

In conclusion, we can compute any $FO + M^2$ formula in $uAHAT_k^0$ by inductively computing its constituent formulas and storing their values over $k$ variables using $n^k$ padding tokens. The result extends to $AHAT_k^0$ by applying Proposition 1. $\qquad\square$

Our construction for Lemma 2 somewhat resembles (though is distinct from) a result of Lange (2004, Corollary 6.8) that any problem in $TC^0$ can be reduced to majority logic via a transformation that appends some extra tokens to the input. However, their transformation is not padding since it appends some "non-blank" tokens, and their result also does not clearly apply to transformers.

Combined with previous results about transformers being in FO-uniform $TC^0$, Lemma 2 yields an exact characterization of constant-depth transformers with padding:

**Theorem 1.** $uAHAT_*^0 = AHAT_*^0 = $ FO-*uniform* $TC^0$, *i.e.,* $FO[M, bit]$.

*Proof.* It is known that $AHAT^0 \subseteq$ FO-uniform $TC^0$ (Merrill and Sabharwal, 2023a,c; Chiang, 2025), and this generalizes to transformers with padding tokens (Pfau et al., 2024). We will show FO-uniform $TC^0 \subseteq uAHAT_*^0$. Mix Barrington et al. (1990, Proposition 10.3) proved that FO-uniform $TC^0$ is definable by FO formulas with $M^2$ quantifiers: notably, the bit predicate is not necessary when using $M^2$ quantifiers. Thus, Lemma 2 establishes that FO-uniform $TC^0 \subseteq uAHAT_*^0$. Finally, from Proposition 1, we have $uAHAT_*^0 \subseteq AHAT_*^0$. Hence, $uAHAT_*^0 = AHAT_*^0 = $ FO-uniform $TC^0$. $\qquad\square$

A technical hurdle to obtaining this characterization in prior work was simulating the bit predicate in standard definitions of $TC^0$ (Pfau et al., 2024). Our results circumvent this by instead relating transformers to $FO + M^2$, which is equivalent to standard $FO[M, bit]$ (Mix Barrington et al., 1990).

It thus follows from Theorem 1 that padded transformers can also simulate bit as well as all of FO, which will be useful in the following section for simulating FO reductions.

Lastly, we note that the arguments leading to Theorem 1 work for both logarithmic and polynomial precision (cf. Appendix A) transformers. This shows that these two levels of precision lead to identical power for fixed-depth padded transformers.

## 5  Log$^d$-Looped Padded Transformers Recognize Exactly FO-uniform TC$^d$

The notion of *completeness* of a problem (or language) for a complexity class under certain types of *reduction* has played a key role in computational complexity. Just like it has been useful for reasoning about the expressivity of resource-bounded standard models of computation (Turing machines, circuit models, etc.), we show it can also be used to reason about the expressivity of padded transformers.

We begin by formally defining the notion of reductions in terms of predicates or languages (rather than string-to-string functions). This will make it easier to precisely implement reductions inside transformers, which produce contextual representations of prefixes of the input string in parallel, in contrast to the more standard definition of reductions as string-to-string functions.

**Definition 9.** Let $b(i)$ be the binary encoding of $i \in \mathbb{N}$ in some alphabet $\Sigma \supseteq \{0, 1\}$. Let R be a class of languages. We say a transduction $f : \Sigma^* \to \Sigma^*$ is an R *reduction* if $|f(w)|$ is polynomial in $|w|$ and the language $R_f = \{(w, b(i), \sigma) \mid f(w)_i = \sigma\}$ is in R.[3]

Definition 9 recovers the standard notions of FO and L reductions. A transformer can be said to compute a reduction $f$ if it recognizes the language of triples $(w, i, \sigma)$ defined above. Since $|\Sigma|$ is finite and transformers can equally easily output a 'token' in $\Sigma$ instead of just 0/1, it is in fact more natural to require the transformer to compute a functional form of this language, namely compute $r_f(w, i)$ defined as $f(w)_i$. Our constructions work under both views, though the latter is often more natural and efficient. Formally:

**Definition 10.** We say that a *transformer computes an* R *reduction* $f$ if it either recognizes the language $R_f = \{(w, i, \sigma) \mid f(w)_i = \sigma\}$ in R or computes the function $r : \Sigma \times \mathbb{N} \to \Sigma$ defined as $r_f(w, i) = f(w)_i$, where, in either case, $i$ is encoded in binary.

**Lemma 3.** *Let* C, R *be classes of languages. Let language* $L$ *be* C*-complete under* R *reductions. If* AHAT$^d_*$ *transformers can recognize* $L$ *and compute every* R *reduction, then* C $\subseteq$ AHAT$^d_*$.

*Proof Sketch.* Consider any language $L' \in$ C. By the assumed completeness of $L$, there exists an R reduction $f$ that maps inputs of $L'$ into inputs of $L$ such that $w \in L'$ if and only if $f(w) \in L$. From the last precondition of the theorem, there exists a causally masked log$^d$-depth padded transformer $T_f$ that recognizes the corresponding reduction language $R_f$ or, equivalently, computes the reduction function $r_f$ (cf. Definition 10). We will assume the latter, i.e., that $T_f$ computes $r_f$, though the construction can also be made to work if $T_f$ checks membership in $R_f$. Additionally, we also have from a precondition that there exists a causally masked log$^d$-depth padded transformer $T_L$ that recognizes $L$.

The idea is to "stack" $T_L$ on top of $T_f$ to obtain a log$^d$-depth padded transformer $T$ that recognizes $L'$. Intuitively, given an input $w$, the first set of layers of $T$ will compute $f(w)$ tokenwise, by computing $r(w, i) = f(w)_i$ for every $i$ in parallel. To this end, we will essentially make $|f(w)|$ copies of the padding tokens needed by $T_f$ and perform the computation of $T_f$ independently for each $f(w)_i$. The second and final set of layers of $T$ will then check whether the string $f(w)$ produced by the first set of layers is in $L$, which will hold if and only if $w \in L'$. See full proof in Appendix C.  □

Combined with results in prior work, it follows from Lemma 3 (see proof below) that padded log-depth transformers can recognize any language in NL:

**Lemma 4.** NL $\subseteq$ AHAT$^1_*$.

*Proof.* Let $L$ be the graph connectivity problem, class C be NL, and class R be FO. We will show that the preconditions of Lemma 3 are met, from which it will follow that NL $\subseteq$ AHAT$^1_*$. First, graph

---

[3]Here $f(w)_i$ denotes the $i$-th token of $f(w)$ if $i \leq |f(w)|$, and a special symbol $\square \notin \Sigma$ otherwise.

connectivity is known to be NL-complete under FO reductions (Immerman, 1998). Second, Merrill and Sabharwal (2025) recently showed that mixed-masked log-depth transformers with cubic padding can recognize the graph connectivity problem $L$, i.e., $L \in \mathsf{mAHAT}^1_*$. From Proposition 1, it follows that $L \in \mathsf{AHAT}^1_*$. Finally, it follows from Theorem 1 that fixed-depth causally masked transformers can recognize languages in $\mathsf{FO}[\mathsf{M}, \mathsf{bit}]$, and hence also languages in $\mathsf{FO}$. Such transformers can therefore compute $\mathsf{FO}$ reductions $f$ in the sense of Definition 10, i.e., compute the function $r_f(w, i)$ defined as $f(w)_i$, the $i$-th bit of $f(w)$. Thus, Lemma 3 applies. $\qquad\square$

Since L reductions are in NL, we can bootstrap this result to obtain the following stronger result. For this, we will leverage the notion of reductions and the completeness of the problem of evaluating a given "wide" $\log^d$ depth circuit, formalized in Definition 15 (Appendix D).

**Lemma 5.** *For $d \geq 0$, FO-uniform* $\mathsf{TC}^d \subseteq \mathsf{AHAT}^d_*$.

*Proof.* We will apply Lemma 3 with the wide-$\mathsf{TC}^d$ circuit evaluation problem (Appendix D) as $L$, FO-uniform $\mathsf{TC}^d$ as class C, and L as class R. We next argue that the preconditions of Lemma 3 are met, which will finish the proof. First, Corollary 13.1 (Appendix D.2) shows that $\log^d$-depth looped transformers (without padding) can solve the wide-$\mathsf{TC}^d$ circuit evaluation problem, i.e., $L \in \mathsf{AHAT}^d_0$. Second, Corollary 14.2 (Appendix D.3) shows that $L$ is complete for FO-uniform $\mathsf{TC}^d$ under L reductions. Finally, Lemma 4 implies that $\log^d$-depth transformers for $d \geq 1$ can recognize any language in L, and thus compute any L reduction in the sense of Definition 10. Applying Lemma 3, we conclude that FO-uniform $\mathsf{TC}^d \subseteq \mathsf{AHAT}^d_*$. $\qquad\square$

The proof heavily leverages the fact that wide-$\mathsf{TC}^d$ circuit evaluation $\mathsf{TC}^d$-complete, which we show in Appendix D.1. To our knowledge, formalizing this $\mathsf{TC}^d$-complete problem (or, in fact, any natural $\mathsf{TC}^d$-complete problem) is a novel contribution. We next note the following extension of a known result about fixed-depth transformers. See Appendix C for a proof, which leverages the fact that recurrent composition of poly-size FO-uniform circuit families remains FO-uniform (Lemma 9, Appendix B.1):

**Lemma 6.** *For $d \geq 1$,* $\mathsf{AHAT}^d_* \subseteq$ *FO-uniform* $\mathsf{TC}^d$.

Combining Lemmas 5 and 6, we obtain an exact characterization of $\mathsf{AHAT}^d_*$ for $d \geq 1$:

**Theorem 2.** *For any $d \geq 1$,* $\mathsf{AHAT}^d_* =$ *FO-uniform* $\mathsf{TC}^d$.

Taking the union over all $d$, we obtain $\bigcup_{d=0}^{\infty} \mathsf{AHAT}^d_*$ as the class of languages recognized by *polylogarithmic-looped padded transformers*. Theorems 1 and 2 imply that this class is the same as $\bigcup_{d=0}^{\infty}$ FO-uniform $\mathsf{TC}^d$, which in turn is the same as the class FO-uniform NC, where NC is the class of all "parallelizable" languages—those recognized by polylogarithmic depth (and polynomial size) circuit families. We therefore have:

**Corollary 2.1.** *Polylog-looped poly-padded transformers recognize exactly* FO-*uniform* NC.

As before, we note that the arguments leading to Theorem 2 and Corollary 2.1 work for both logarithmic and polynomial precision (cf. Appendix A) transformers. This shows that these two levels of precision lead to identical power for polylog-looped polynomially-padded transformers.

## 6 Uniformity Collapse for Circuit Classes

Our results on looped transformers involved substantial analysis of uniform polylogarithmic-depth circuit classes. This analysis led us to prove a novel result about uniform circuit families, which both slightly strengthens our results about looped transfomers and may be of independent interest.

The result concerns the strength of different uniformity conditions for circuit families, characterizing conditions under which variants of circuit classes with different uniformity conditions *collapse* to express the same class of languages (Proposition 6 in Appendix E). Applying this general result to (wide) $\mathsf{TC}^d$ circuits, we show that for $d \geq 1$, both NL-uniformity and L-uniformity collapse to the weaker notion of FO-uniformity for both $\mathsf{AC}^d$ and $\mathsf{TC}^d$ circuits:

**Theorem 3** (Uniformity Collapse). *For any $d \geq 1$, the following equivalences hold:*

$$\mathsf{FO}\text{-}\textit{uniform } \mathsf{AC}^d = \mathsf{L}\text{-}\textit{uniform } \mathsf{AC}^d = \mathsf{NL}\text{-}\textit{uniform } \mathsf{AC}^d$$
$$\mathsf{FO}\text{-}\textit{uniform } \mathsf{TC}^d = \mathsf{L}\text{-}\textit{uniform } \mathsf{TC}^d = \mathsf{NL}\text{-}\textit{uniform } \mathsf{TC}^d.$$

The key idea, formalized in Appendix E, is to leverage the fact that NL and L themselves are in FO-uniform $\mathsf{AC}^d$ for $d \geq 1$, and therefore the NL or L machine that builds a circuit family can itself be simulated in FO-uniform $\mathsf{AC}^d$ as well. Thus all one needs to do is compose this "circuit building" circuit family with a "circuit evaluation" circuit family. To this end, we show that functions computed by FO-uniform circuit families are closed under fixed compositions (Proposition 3).

Theorem 3 implies that, for $d \geq 1$, $\mathsf{AHAT}_*^d$ recognizes not just FO-uniform $\mathsf{TC}^d$ but also L-uniform $\mathsf{TC}^d$ because these classes are the same.

# 7    Conclusion

Our results in this work give a precise theoretical understanding of how padding and looping—two ways to dynamically expand the computational resources of a transformer at inference time—increase the expressive power of transformers. Padding expands the circuit *width* of transformers, allowing them to resolve logical formulas over more variables. As a consequence of this, polynomially padded transformers can recognize *exactly* $\mathsf{TC}^0$, which was previously known only as an upper bound (Merrill and Sabharwal, 2023a; Pfau et al., 2024). In contrast, looping increases the *depth* of transformers. Applying looping on top of padding, we extended our result to show that $\log^d$-depth, padded transformers recognize exactly $\mathsf{TC}^d$. This means that transformers with polynomial padding and polylogarithmic looping converge to recognizing NC, the largest class of problems that can be solved with parallel computation. In contrast, transformers with CoT have greater expressive power under standard complexity conjectures (Merrill and Sabharwal, 2023b), but suffer from slow sequential decoding. Thus, while looping and padding are not as powerful as CoT, our results suggest they are quite effective ways to expand transformers' expressive power while preserving parallelism.

Several interesting open questions remain from this work. On the theoretical side, it would be valuable to develop a more finegrained characterization of $\mathsf{AHAT}_k^d$, i.e., looped transformers where the padding is at most $O(n^k)$ for some *fixed $k$*. On the empirical side, while looped and padded transformers have both been explored already to an extent, it would be interesting to see whether these approaches could be successfully integrated to improve the performance of transformers on hard reasoning tasks. Further developing these approaches could ultimately lead to inference-time compute methods based on padding and looping that increase transformers' expressive power without sacrificing parallelism, providing a more efficient alternative to CoT for solving moderately parallelizable problems.

# Limitations

**Practical Concerns for Looped Transformers with Padding.** In this paper, we showed looped transformers with padding are quite *expressive*. However, the degree to which transformers can learn to use looped layers is an important open question. In particular, practical details might be important here, such as an appropriate parameterization where features can be learned in later layers (Dey et al., 2025). Another practical caveat is that, while padding is efficiently parallelizable, it will extend the context length, incurring more memory overhead for the forward pass.

**Transformer Model.** Here we have assumed AHATs here rather than softmax-attention transformers (SMATs). For any fixed maximum context length, it is possible scale the temperature of SMAT heads to arbitrarily approximate AHAT heads, but for unbounded context lengths, SMATs may not be able to simulate AHATs. We thus view the AHAT as mild simplification of the SMAT that abstracts away issues with soft attention for simulating hard attention over very long contexts. It would also be interesting to better understand the necessity of the masked pre-norm assumption.

## Acknowledgments

We appreciate discussions with Selim Jerad, Andy Yang, Michael Cadhilac, and attendees of the Formal Languages and Neural Networks (FLaNN) seminar. This project was supported by the National Science Foundation (NSF) through award 1922658 and WM's NSF Graduate Research Fellowship. WM was also supported by a Two Sigma PhD Fellowship and the Allen Institute for AI.

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

## A  Datatype Assumptions

We adapt the $p$-precise datatype model from Merrill and Sabharwal (2025). We encode scalars as strings in $\{0,1\}^p$, where $p$ can be a function of $n$. If $p$ depends on $n$, activations, but not models parameters, can depend on $n$. A *datatype* $\mathbb{D}_p$ assigns a numerical semantics for each string in $\{0,1\}^p$. For $x \in \mathbb{R}$, let $[x]_{\mathbb{D}_p}$ be $x$ rounded into $\mathbb{D}_p$, i.e., the bitstring whose numerical value in $\mathbb{D}_p$ is closest to $x$ (breaking ties in favor of the higher value). We define our datatype $\mathbb{D}_p$ to satisfy the following:

**Definition 11** (*p*-Precise Operations)**.** Let $f : \mathbb{R}^k \to \mathbb{R}$ be an operation with $p$-precision realization $\tilde{f} : \mathbb{D}_p^k \to \mathbb{D}_p$. We say $\tilde{f}$ is $p$-precise if, for any $x_1, \ldots, x_k \in \mathbb{R}$ exactly representable in $\mathbb{D}_p$,

$$[f(x_1, \ldots, x_k)]_{\mathbb{D}_p} = \tilde{f}([x_1]_{\mathbb{D}_p}, \ldots, [x_k]_{\mathbb{D}_p}).$$

To apply Definition 11, we view the summation in attention heads as an $n$-ary operation. We also view layer-norm as a single operation from $\mathbb{R}^m \to \mathbb{R}^m$. Lastly, we assume that these operations in the computation graph are defined in the standard way and are thus computable in $\mathsf{TC}^0$ (Merrill and Sabharwal, 2023a,c).

We consider two natural instantiations of $\mathbb{D}_p$ in the main text: *log precision*, with $p = c \log n$ (Merrill and Sabharwal, 2023a), and *polynomial precision*, with $p = n^c$ (Chiang, 2025), for some fixed $c > 0$. All our results go through with either datatype, showing their equivalence for polylogarithmically-looped polynomially-padded transformers.

## B  Uniformity of Circuit Classes

To make our arguments rigorous for highly uniform (i.e., FO-uniform) circuit families, it will be necessary to work with a detailed definition of uniformity. For this, we start with a standard formal definition of the *connection language* describing the gates and wires of a circuit family. Here we allow each circuit to have more than one (numbered) output gates.

**Definition 12** (Connection Language; cf. Vollmer, 1999)**.** Define the connection language for a circuit family $\mathcal{C} = \{C_n\}_{n=0}^{\infty}$ as $L_{\mathcal{C}} = L_{\text{gate}} \cup L_{\text{wire}}$ where

$L_{\text{gate}} = \{a^n igk\ell \mid \text{gate } i \text{ of } C_n \text{ is type } g, \text{ is } k\text{-th input gate if } k \geq 0, \text{ and is } \ell\text{-th output gate if } \ell \geq 0\}$
$L_{\text{wire}} = \{a^n ij \mid C_n \text{ has a wire } i \to j\}$

and $i, j \in \mathbb{Z}_{\geq 0}$, $g$ is a gate type, and $k, \ell \in \mathbb{Z}_{\geq -1}$. Here $i, j, k, \ell$ are represented using exactly $c \log n$ bits for some fixed $c \in \mathbb{Z}_{\geq 1}$.

We will assume that gates in all circuits are numbered so that there is a block of input gates, followed by a block of intermediate gates, followed by a block of output gates. Thus, there exist specific gate thresholds that denote the boundaries for input and output gates.

**Definition 13** (Generalized Uniformity)**.** A family of circuits $\mathcal{C} = \{C_n\}_{n=0}^{\infty}$ is A-uniform if $L_{\mathcal{C}} \in \mathsf{A}$.

When $\mathsf{A} = \mathsf{L}$, this generalized uniformity notion is equivalent to the standard definition of uniformity in terms of being able to serialize $C_n$ as a function of $1^n$:

**Proposition 2.** $\mathcal{C} = \{C_n\}_{n=0}^{\infty}$ *is* $\mathsf{L}$-*uniform if and only if* $1^n \mapsto \langle C_n \rangle$ *is computable in log space.*

*Proof.* To use the connection language notion of uniformity to serialize the circuit with log space, we simply maintain a counter of our current gate position and edge and call an $\mathsf{L}$ oracle for the connection language to print each gate and wire. In the other direction, we modify the routine that serializes the circuit to only output gate $i$ or edge $i, j$, and we use this to recognize $L_{\mathcal{C}}$. $\square$

### B.1  Uniformity Composition Closures

Our later results about uniform circuit families will rely on the property (formalized below) that for strong-enough classes A, any fixed composition of A-uniform circuit families is also A-uniform. Note that any fixed composition of functions can be decomposed into a fixed number of *serial* and *parallel* compositions of two functions, defined as follows:

**Definition 14** (Function Composition). For functions $f, g \in \{0,1\}^* \to \{0,1\}^*$, their *serial composition* is the function $(g \circ f)(w) = g(f(w))$ and their parallel composition is the function $\langle f, g \rangle(w) = f(w) \cdot g(w)$.

For L-uniform functions, serial and parallel composition clearly preserves L-uniformity, as we can maintain a counter to reroute inputs and outputs between subcircuits appropriately. The same applies to any class A containing L. For FO uniform circuit classes, closure under composition requires a bit more work to justify, as shown in the following two lemmas.

**Lemma 7.** *If functions $f$ and $g$ have polynomial-size* FO-*uniform circuit families, then so does their parallel composition $\langle f, g \rangle$.*

*Proof.* Let $\mathcal{C}^f = \{C_n^f\}_{n=0}^\infty$ be an FO-uniform circuit family for $f$, with connection language $L_f = L_{\mathcal{C}_f} \in$ FO. Similarly, define $\mathcal{C}^g, C_n^g$, and $L_g$ for $g$.

We will decide the connection language $L_{\langle f,g \rangle}$ by querying $L_f$ and $L_g$ to build $C_n^f$ and $C_n^g$ in parallel. Without loss of generality, we assume all gate indices in the input word for a gate query fit into one of the following three cases: (1) at most the number of gates $s_f$ in the circuit $C_n^f$, (2) larger than $s_f$ but at most $s_f + s_g$ where $s_g$ is the number of gates in $C_n^g$, or (3) larger than $s_f + s_g$ but at most $s_f + s_g + o_f$, where $o_f$ is the number of output gates of $C_n^f$. If all gates are at most $s_f$, we simply query $L_f$. If all gates are larger than $s_f$ but at most $s_f + s_g$, we subtract $s_f$ from each gate index and query $L_g$. Finally, if we are querying new gates from $s_f + s_g < i \le s_f + s_g + o_f$, we design the connection language to represent identity gates. Importantly, addition and inequality checks can be performed in FO, and the connection languages $L_f$ and $L_g$ can be queried in FO by construction.

The logic for edge queries in the connection language modifies gate indices according to the same logic as the gate queries with a few exceptions. First, for an edge $(i, j)$, if $i$ is an input for $C_n^f$ (and $C_n^g$) and $i = j - s_f$, we return 1. This ensures that $C_n^f$ and $C_n^g$ both receive the same input. Second, if $i$ is an output for $C_n^f$ and $s_f + s_g < j \le s_f + s_g + o_f$, we also return 1. This copies over the outputs of $C_n^f$ so that they appear at the very end of the combined circuit.

Thus, this connection language constructs a circuit family that computes $f$ and $g$ in parallel and returns their outputs $\langle g(w), f(w) \rangle$. Without loss of generality, their order can be easily permuted to obtain the output $\langle f(w), g(w) \rangle$ of parallel composition. It is clear from the construction that the size of the resulting circuit family is linear in the sizes of $\mathcal{C}^f$ and $\mathcal{C}^g$, and thus polynomial in $n$. $\square$

**Lemma 8.** *If functions $f$ and $g$ have polynomial-size* FO-*uniform circuit families, then so does their serial composition $g \circ f$.*

*Proof.* Let $\mathcal{C}^f = \{C_n^f\}_{n=0}^\infty$ be an FO-uniform circuit family for $f$, with connection language $L_f = L_{\mathcal{C}_f} \in$ FO. Similarly, define $\mathcal{C}^g, C_n^g$, and $L_g$ for $g$.

We will decide the connection language $L_{g \circ f}$ by querying $L_f$ to construct $C_n^f$, then modifying the inputs to $L_g$ to build $C_m^g$ on the output of $f$, which has size $m$. The cases are the same as for parallel composition, except for a few changes. First, we build $C_m^g$ instead of $C_n^g$. Second, when querying input gates for $C_m^g$, we add a condition that routes from the *outputs* of $C_n^f$ rather than its inputs. Finally, we do not construct additional gates to copy over the outputs from $C_n^f$. Thus, this connection language builds a circuit family that computes $g$ on the output of $f$. Similar to the parallel composition case, it is clear from the construction that the size of the resulting circuit family is linear in the sizes of $\mathcal{C}^f$ and $\mathcal{C}^g$, and thus polynomial in $n$. $\square$

Combining Lemmas 7 and 8 along with the fact that any fixed composition of functions can be decomposed into a fixed number of serial or parallel compositions of two functions at a time, we obtain that fixed function composition preserves FO-uniformity of circuit families:

**Proposition 3** (Composition Preserves FO-Uniformity). *Any fixed composition of polynomial-size* FO-*uniform circuit families is also polynomial-size and* FO-*uniform.*

**Lemma 9** (Recurrent Composition). *Let $m(n)$, $d(n)$, and $r(n)$ be functions at most polynomial in $n$, with $m(n)$ and $r(n)$ definable as variables in* FO *given $a^n$. Let $f : \{0,1\}^{m(n)} \to \{0,1\}^{m(n)}$ have a polynomial-size* FO-*uniform circuit family with depth $d(n)$. Then the function $f^{r(n)}$ (i.e., $f$*

*called recurrently on itself $r(n)$ times) has an* FO-*uniform circuit family of polynomial size and depth* $d(n)\ r(n)$.

*Proof.* Let $\mathcal{C}^f = \{C_n^f\}_{n=0}^\infty$ be a polynomial size FO-uniform circuit family for $f$, with connection language $L_f$. Without loss of generality, we assume the number of gates in $C_n^f$ is $2^{s(n)}$ for some integer $s(n)$ (this can always be achieved by padding $C_n^f$ if necessary, in a way that increases size by at most a factor of 2). Since the circuit family is of polynomial size, $s(n) = \mathrm{O}(\log n)$ and can be computed in FO as the smallest $j$ s.t. $\mathrm{bit}(s_f, j) = 1$, where $s_f$ is the number of gates in $C_n^f$. We will construct an FO-uniform circuit family $\mathcal{C} = \{C_n\}_{n=0}^\infty$ where $C_n$ consists of $f(n)$ iterations of $C_n^f$. By construction, $C_n$ has size at most polynomial and depth $d(n)\ r(n)$.

The remainder of the proof will justify that $\mathcal{C}$ is FO-uniform by defining its connection language $L_{\mathcal{C}}$ in terms of $L_f$, starting with gate queries. Given a gate query $w$ with index $i$, we first compute $r(n)$, which is possible by construction. We then compute $S = r(n) \cdot 2^{s(n)}$ by left shifting $r(n)$ by $s(n)$, and, if $i \geq S$, we reject $w$. Otherwise, we compute $i' = i \mod 2^{s(n)} = \mathrm{bit}(i, s(n))$, which can be computed in FO by reading the $s(n)$-th bit from the start of $i$ in $w$. We then compute $u$ as a new query where $i$ is replaced by $i'$. We then query whether $u \in L_f$. This ensures that $C_n$ repeats all the gates in $C_n^f$ stacked in $r(n)$ blocks.

Given an edge query $w$ between $(i, j)$, we first compute $i' = i \mod 2^{s(n)}$ and $j' = j \mod 2^{s(n)}$ as above. Additionally, let $q_i = \lfloor i/2^{s(n)} \rfloor$ and $q_j = \lfloor j/2^{s(n)} \rfloor$. We compute $q_i$ and $q_j$ in FO by right-shifting by $s(n)$. If $q_i = q_j$, we follow similar logic to the gate case, constructing $u$ by replacing $i$ with $i'$ and $j$ with $j'$. We then query whether $u \in L_f$. This has the effect of constructing all edges within a block of $C_n$ analogously to those in $C_n^f$. Additionally, if $q_i + 1 = q_j$, $i' \geq s_f - m(n)$, and $j' \leq m(n)$, we return a 1 for this edge; recall that $m(n)$ is the number of inputs as well as the number of outputs of $f$. This has the effect of routing the output of each block as the input for the next block. Thus, $\mathcal{C}$ computes $f^{r(n)}$ and is FO-uniform. $\qquad\square$

## C  Omitted Proofs

**Proposition 1.** *Unmasked (with position encoding $1/i$ or $i/n$; cf. Lemma 10 in §C) and mixed-masked padded transformers (i.e., uAHATs and mAHATs) can be simulated by causally masked transformers (AHATs). Specifically, for any $d, k \in \mathbb{N}$, the following holds for the corresponding problem classes:*

1. $\mathsf{uAHAT}_k^0 \subseteq \mathsf{mAHAT}_k^0 \subseteq \mathsf{AHAT}_{\max\{k,1\}}^0$
2. $\mathsf{uAHAT}_k^d \subseteq \mathsf{mAHAT}_k^d \subseteq \mathsf{AHAT}_{1+\max\{k,1\}}^d$ *for* $d \geq 1$
3. $\mathsf{uAHAT}_*^d \subseteq \mathsf{mAHAT}_*^d \subseteq \mathsf{AHAT}_*^d.$

*Proof.* As noted in the proof of Lemma 1, the $1/i$ position encoding used in our unmasked transformers can be computed by causally masked transformers, and hence also by mixed-masked transformers. Thus, unmasked padded looped transformers with position encoding $1/i$ constitute a special case of mixed-masked padded looped transformers, and can thus be trivially simulated by the latter. We will next describe how to leverage Lemma 1 to convert an unmasked padded looped transformer to a masked one, on a per-head basis. The construction will leave the computation of masked heads unchanged, making the approach suitable for converting both unmasked and mixed-masked padded transformers to masked ones.

Let $E$ be an unmasked looped encoder in $\mathsf{uAHAT}_k^d$. Then $E$ has depth $\ell = O(\log^d n)$ and operates over $n + O(n^k)$ tokens (including original input and padding tokens). By Lemma 1, $E$ can be simulated by a causally masked decoder $D$ with depth $\ell$ and with $\ell \cdot (n + O(n^k))$ new padding tokens (after the original $\mathrm{O}(n^k)$ padding tokens used by $E$). Thus, the total number of padding tokens $D$ uses is $p = O(n^k) + \ell \cdot (n + O(n^k))$. Since $\ell = O(\log^d n)$, this simplifies to $p = O(\log^d(n) \cdot n^{k'})$ where $k' = \max\{k, 1\}$. This is $\mathrm{O}(n^{k'})$ when $d = 0$, and $\mathrm{O}(n^{k'+1})$ when $d \geq 1$. This finishes the proof of the first two parts. For the third part, observe that by definition, $\mathsf{uAHAT}_*^d = \bigcup_{k=0}^\infty \mathsf{uAHAT}_k^d$. From the above, this in turn is contained in $\bigcup_{k=0}^\infty \mathsf{AHAT}_{k'+1}^d \subseteq \mathsf{AHAT}_*^d$, as desired.

For the mixed-masking case, the construction in Lemma 1 will preserve the original computation of causally masked heads if we add an additional term with large negative weight $C$ (fixed w.r.t. $n$) that is activated if the index within the block is greater than $b_i$. We set $C$ large enough to dominate all other terms in the inner product computation at each token. As a result, the head is constrained to only attend to previous tokens within the block. Applying this modified construction on a case-by-case basis per head, we can take this proposition to apply equally to transformers with mixed-masking. $\square$

The above simulation of unmasked padded looped transformers (with position encoding $1/i$) with mixed-mask transformers can, in fact, be extended even to the case where the unmasked transformer uses $i/n$ positional encoding:

**Lemma 10.** *There exists a mixed-mask sublayer with one masked head and one unmasked head that computes $\phi(i/n)$.*

*Proof.* We use two attention heads, one causally masked and one unmasked, both of which attend uniformly with value 1 only for the beginning-of-sequence symbol. The first head thus computes $1/i$, while the second head computes $1/n$. We can then combine these values to compute $\phi(1/n, 1/i) = \phi(i/n)$. $\square$

**Lemma 3.** *Let* $\mathsf{C}, \mathsf{R}$ *be classes of languages. Let language $L$ be* $\mathsf{C}$-*complete under* $\mathsf{R}$ *reductions. If* $\mathsf{AHAT}_*^d$ *transformers can recognize $L$ and compute every* $\mathsf{R}$ *reduction, then* $\mathsf{C} \subseteq \mathsf{AHAT}_*^d$.

*Proof.* Consider any language $L' \in \mathsf{C}$. By the assumed completeness of $L$, there exists an $\mathsf{R}$ reduction $f$ that maps inputs of $L'$ into inputs of $L$ such that $w \in L'$ if and only if $f(w) \in L$. From the last precondition of the theorem, there exists a causally masked $\log^d$-depth padded transformer $T_f$ that recognizes the corresponding reduction language $R_f$ or, equivalently, computes the reduction function $r_f$ (cf. Definition 10). We will assume the latter, i.e., that $T_f$ computes $r_f$, though the construction can also be made to work if $T_f$ checks membership in $R_f$. Additionally, we also have from a precondition that there exists a causally masked $\log^d$-depth padded transformer $T_L$ that recognizes $L$.

The idea is to "stack" $T_L$ on top of $T_f$ to obtain a $\log^d$-depth padded transformer $T$ that recognizes $L'$. Intuitively, given an input $w$, the first set of layers of $T$ will compute $f(w)$ tokenwise, by computing $r(w, i) = f(w)_i$ for every $i$ in parallel. To this end, we will essentially make $|f(w)|$ copies of the padding tokens needed by $T_f$ and perform the computation of $T_f$ independently for each $f(w)_i$. The second and final set of layers of $T$ will then check whether the string $f(w)$ produced by the first set of layers is in $L$, which will hold if and only if $w \in L'$. We next make this idea more concrete.

Compute Reduction. Suppose $T_f \in \mathsf{AHAT}_k^d$ and let $n = |w|$. Then, on input $(w, i)$, where $i$ is represented in binary, $T_f$ uses $\mathrm{O}(n^k)$ padding tokens to compute $f(w)_i$, which we upper bound by $n^{k+1}$ for sufficiently large $n$ (for small $n$, we assume $T_f$ instead uses a fixed lookup table (and no padding). There is a uniform $\mathsf{TC}^0$ circuit family that computes $\mathrm{bit}(i, j)$ from input $\phi(i), \phi(j)$, so, by Theorem 1, there exists a transformer $T_{\mathrm{bit}}$ that computes bit with $n^b$ padding tokens, for some $b$ and sufficiently large $n$ (again, for smaller $n$, we assume $T_{\mathrm{bit}}$ uses a fixed lookup table). Since $|f(w)|$ is bounded by some polynomial $n^c$, we know that any $i \leq |f(w)|$ takes at most $c \log n$ bits to specify, which we upper bound by $c' = n$ (since it's unclear how a transformer would compute $\log n$).

We will construct $n^c$ blocks of padding tokens, each of size $B = c'n^b + n^{k+1}$. Block $i$ will first compute the binary expansion of $i$ in its first $c'n^b$ padding tokens using $T_{\mathrm{bit}}$, and then use the remaining $n^{k+1}$ padding tokens to consume $(w, i)$ using $T_f$ and return $r_f(w, i)$. At each token $t$ in block $i$, we compute $\phi(i)$ as $\phi(\lfloor t/B \rfloor)$, using the fact that we can compute integer division with a fixed block of transformer layers (Merrill and Sabharwal, 2025) and having computed $B$ as a function of $n$ in earlier layers (since it is in $\mathsf{TC}^0$). For $t' \leq c'$, token $t = t'n^b$ computes $\phi(t')$ via division similarly to $\phi(i)$ and then $\mathrm{bit}(i, t')$. This recovers the binary representation of $i$ stored across tokens $t = t'n^b$ in block $i$, for $1 \leq t' \leq c'$. Finally, we apply a slightly modified $T_f$ over the block. Specifically, we add a new term to each attention head so that it only attends over the input tokens and some tokens within the current block: those satisfying $t = t'n^b$ for some $1 \leq t' \leq c'$ or $t > c'n^b$: since $c' = n$, these predicates are simple to check. As a result, block $i$ simulates $T_f$ over $w$ with $n^k$ padding. Thus, the final token of block $i$ computes $r_f(w, i) = f(w)_i$.

Solve Complete Problem. We will use additional layers to check whether $f(w) \in L'$. We are given that there exists a transformer $T_L$ that, on input $w'$, checks whether $w' \in L'$. For each attention head in $T_L$, we add a new term to the attention score that is very negative if that token is not the final token in some attention block. Thus, each head in $T_L$ will only attend over tokens that are final in some block. We also modify $T_L$ so that it uses $\phi(i)$ in place of the position embedding for token $t$. Thus, $T_L$ computes whether $f(w) \in L'$, which is equivalent to recognizing whether $w \in L$. □

**Lemma 6.** *For $d \geq 1$, $\mathsf{AHAT}_*^d \subseteq$ FO-uniform $\mathsf{TC}^d$.*

*Proof.* Let $L$ be a language in $\mathsf{AHAT}_*^d$ and let $T$ be a looped, padded AHAT transformer that, when unrolled $c \log^d n$ times for large enough $n$,[4] recognizes whether $w \in L$ for any input $w$ with $|w| = n$. Let $\langle A, B, C \rangle$ be the partition of layers of $T$ where $A$ is the set of initial layers, $B$ is the block that's repeated $c \log^d n$ times on inputs of length $n$, and $C$ is the set of final layers. Each of these itself is a fixed-depth padded transformer; let's call these $T_A$, $T_B$, and $T_C$. By prior results (Merrill and Sabharwal, 2023a,c; Chiang, 2025), there are FO-uniform $\mathsf{TC}^0$ circuit families $\{C_n^A\}_{n=0}^\infty$, $\{C_n^B\}_{n=0}^\infty$, and $\{C_n^B\}_{n=0}^\infty$ that simulate transformers $T_A, T_B$, and $T_C$, respectively. Let $T_R$ be $T_B$ iterated $\log n$ times, and Let $T_R^d$ be $T_B$ iterated $\log^d n$ times. We justify that $r(n) = \lceil \log n \rceil$ is definable as a variable in FO given $a^n$: compute the index of the last $a$ and then find the greatest bit index that is 1. We now invoke Lemma 9 to show that $T_R$ has an FO-uniform $\mathsf{TC}^d$ circuit family; we can repeat this process $d$ times to get an FO-uniform $\mathsf{TC}^d$ circuit family for $T_R^d$. By Lemma 8, there is a also an FO-uniform $\mathsf{TC}^d$ circuit family that computes the serial composition of $T_A, T_R, \ldots, T_R$, and $T_C$, where $\ldots$ accounts for a fixed repetition of $T_R$ to account for a constant $c$ on the depth $c\lceil \log n \rceil^d$. Thus, we conclude that $L \in$ FO-uniform$\mathsf{TC}^d$. □

# D   Wide-$\mathsf{TC}^d$ Circuit Evaluation Problem

To formalize the circuit evaluation problem, we will use the following serialized format for representing a circuit. This format is a simplification of the one used by Merrill and Sabharwal (2023a), with two main differences: (a) instead of a single threshold gate, we use AND, OR, NOT, and MAJ (the majority gate), which will simplify the description of the construction; and (b) we do not require the gates in the serialization to be sorted in any particular order, which makes it easier (and perhaps even possible) to have a log-space reduction from any L-uniform $\mathsf{TC}^d$ language to the circuit evaluation problem in this specific serialization format. The syntax of this circuit format is governed by the following grammar:

$$\text{Circuit} \rightarrow \text{Gate}^*$$
$$\text{Gate} \rightarrow \text{X} \mid \text{Op Arg}^*$$
$$\text{Op} \rightarrow \text{AND} \mid \text{OR} \mid \text{NOT} \mid \text{MAJ}$$
$$\text{Arg} \rightarrow \text{\&1}^*$$

Semantically, we take the $k$-th X gate to return the $k$-th input bit. Other gates retrieve the values of the gates referred to by their argument pointers and apply the associated logical function. We take the final gate in the serialization to be the output gate of the circuit. Note that not all strings in this grammar represent a well-formed circuit, but any valid circuit can be serialized in this format.

As an example, the threshold circuit $\text{Majority}(x_1, x_2 \lor x_3, \neg x_3)$ stating that at least two of $x_1$, $x_2 \lor x_3$, and $\neg x_3$ should be true, would be represented as follows:

$$\underbrace{\text{X X X}}_{\text{input}} \ \underbrace{\text{MAJ \&1 \&11111 \&111111}}_{\text{Majority gate}} \ \underbrace{\text{OR \&11 \&111}}_{\text{Or gate}} \ \underbrace{\text{NOT \&111}}_{\text{Not gate}}$$

Note that the non-input gates need not be serialized in this particular order. The following is also an equally valid serialization of the same formula:

$$\underbrace{\text{X X X}}_{\text{input}} \ \underbrace{\text{OR \&11 \&111}}_{\text{Or gate}} \ \underbrace{\text{NOT \&111}}_{\text{Not gate}} \ \underbrace{\text{MAJ \&1 \&1111 \&11111}}_{\text{Majority gate}}$$

We now formalize the circuit evaluation problem, for any class of circuit families, such as the class $\mathsf{TC}^d$ of $\log^d$-depth circuit families:

---

[4]As before, for small $n$, we assume $T$ uses a lookup table.

**Definition 15** (C Circuit Evaluation). Let C be a class of (potentially non-uniform) circuit families. The C circuit evaluation problem is defined as follows:

- Input: $(x, \langle C \rangle)$ where $x \in \{0, 1\}^*$ is a string and $\langle C \rangle$ is the serialization of a circuit $C$ such that $C = C_{|x|}$ for some circuit family $\{C_n\}_{n=0}^{\infty} \in$ C.

- Output: The value $C(x)$.

For example, the case where C = P/poly yields the generic *circuit value problem*, which is known to be P-complete. We focus here to the case of C = $\mathsf{TC}^d$, i.e., the $\mathsf{TC}^d$ *circuit evaluation problem*. It is somewhat intuitive that this problem is hard for the class A-uniform $\mathsf{TC}^d$ as long as A is strong enough to build circuits; we formalize this later in Lemma 11. However, the $\mathsf{TC}^d$ circuit evaluation problem is *not* necessarily in the class $\mathsf{TC}^d$. To see this, suppose to the contrary that there exists an A-uniform $\mathsf{TC}^d$ circuit family $\mathcal{C} = \{C_n\}_{n=0}^{\infty}$ that solves the $\mathsf{TC}^d$ circuit evaluation problem. Then, for large enough $n$, each circuit $C_n \in \mathcal{C}$ has depth upper bounded by $c \log^d n$ for a *fixed* $c > 0$. This $c$ being fixed is problematic—if one were to try to evaluate on string $x$ a circuit $C'_n$ from a $\mathsf{TC}^d$ circuit family that has depth $c' \log^d n$ where $c' > c$ (that is, invoke circuit evaluation for input $(x, \langle C'_n \rangle)$), the intuitive approach would require a circuit whose depth is larger than that of $C_n$.

To address this, we now formalize a *constrained version* of the $\mathsf{TC}^d$ circuit evaluation problem that is, in fact, within the class FO-uniform $\mathsf{TC}^d$. We achieve this via the **wide-$\mathsf{TC}^d$ circuit evaluation problem**, where the class wide-$\mathsf{TC}^d$ is defined as follows:

**Definition 16** (Wide Circuits). Let **wide-$\mathsf{TC}^d$** $\subseteq \mathsf{TC}^d$ be the class of circuit families $\{C_n\}_{n=0}^{\infty}$ such that there exists some $c$ such that, for large $n$, the depth of $C_n$ is at most $c \log^d n$ and, crucially, the size is *at least* $n^c$.

That is, wide-$\mathsf{TC}^d$ enforces that the size (and hence the width) of the circuit is large relative to its depth. In particular, for every wide-$\mathsf{TC}^d$ circuit family $\mathcal{C}$, there is a $c > 0$ such that the circuit $C_n$ has serialization of size $\Omega(n^c)$, and depth at most $c \log^d n$. Thus the depth of $C_n$ is at most $\log^d N$ where $N = n + \Omega(n^c)$ is the overall size of the input $(x, \langle C_n \rangle)$ to the corresponding circuit evaluation problem. As we will show later, this allows the wide-$\mathsf{TC}^d$ circuit evaluation problem to be solved by a transformer (Corollary 13.1) as well as by an FO-uniform $\mathsf{TC}^d$ circuit family (Lemma 14) using precisely $\log^d N$ iterations (manifested as loops of a transformer or FO-uniform $\mathsf{TC}^0$ block, respectively), irrespective of the $\mathcal{C}$-dependent value of $c$.

Since it is a class of circuit families, wide-$\mathsf{TC}^d$ can be constrained by uniformity conditions in the natural way. With some abuse of notation, we will use wide-$\mathsf{TC}^d$ to refer to both the class of circuit families and the complexity class of language classes the circuit families recognize. Interestingly, this minimum size constraint imposed by wide-$\mathsf{TC}^d$ does not weaken it as a language class compared to $\mathsf{TC}^d$:

**Proposition 4.** *For any $d \geq 0$, non-uniform wide-$\mathsf{TC}^d$ = non-uniform $\mathsf{TC}^d$.*

*Proof.* Every wide-$\mathsf{TC}^d$ circuit family is, by definition, also a $\mathsf{TC}^d$ circuit family.

Conversely, suppose $L \in \mathsf{TC}^d$. Then there is a circuit family $\{C_n\}_{n=0}^{\infty}$ recognizing $L$, where the depth of $C_n$ is at most $c \log^d n$ for some $c$ and large enough $n$. Consider a modified circuit family $\{C'_n\}_{n=0}^{\infty}$ where, for each $n$, $C'_n$ is a copy of $C_n$ that, if needed, is padded with dummy gates so that it has size at least $n^c$. This modified circuit family also recognizes $L$ but belongs to wide-$\mathsf{TC}^d$, completing the proof. $\square$

In fact, the above equality holds even for uniform variants of these classes:

**Proposition 5.** *For any $d \geq 0$ and A $\supseteq$ FO, A-uniform wide-$\mathsf{TC}^d$ = A-uniform $\mathsf{TC}^d$.*

*Proof.* The proof follows that of Proposition 4. Every A-uniform wide-$\mathsf{TC}^d$ circuit family is, by definition, also a A-uniform $\mathsf{TC}^d$ circuit family. Conversely, suppose $L \in$ A-uniform wide-$\mathsf{TC}^d$.

Then, we have a a circuit family $\mathcal{C} = \{C_n\}_{n=0}^{\infty}$ recognizing $L$, where the depth of each $C_n$ is at most $c \log^d n$ for some $c$ and large enough $n$. By A uniformity, we have the connection language $L_{\mathcal{C}} \in \mathsf{A}$. We define a circuit family $\mathcal{C}'$ that is $\mathcal{C}$ padded with dummy gates to make the number of gates at least $n^c$. The connection language $L_{\mathcal{C}'}$ defaults to $L_{\mathcal{C}}$ for gates that exist in $\mathcal{C}$ and simply returns a dummy gate for other gate indices up to $n^c$. This additional logic can be implemented in FO, so $L_{\mathcal{C}'} \in \mathsf{A} \supseteq \mathsf{FO}$. Thus, the circuit family $\mathcal{C}'$ recognizes $L$ and is A-uniform. $\qquad\square$

## D.1 Hardness of Wide-$\mathsf{TC}^d$ Circuit Evaluation

**Lemma 11.** *For $d \geq 0$, $\mathsf{TC}^d$ circuit evaluation is hard for $\mathsf{L}$-uniform $\mathsf{TC}^d$ under $\mathsf{L}$-reductions.*

*Proof.* Given any $L \in \mathsf{L}$-uniform $\mathsf{TC}^d$, there exists a log-space Turing machine $T_L$ that constructs a circuit family $\{C_n\}_{n=0}^{\infty}$ that recognizes $L$. We can construct an $\mathsf{L}$-reduction from $L$ to the $\mathsf{TC}^d$ circuit evaluation problem as follows. Given an input $x$ whose membership in $L$ we would like to check, the reduction first copies $x$ to the output. It then uses the log-space Turing machine $T_L$ to build the circuit $C_{|x|}$ and output it in the above serialized format. We thus have a log-space reduction from $x$ to $(x, \langle C_{|x|} \rangle)$. We conclude that $\mathsf{TC}^d$ circuit evaluation is hard for $\mathsf{L}$-uniform $\mathsf{TC}^d$ under $\mathsf{L}$-reductions. $\qquad\square$

**Lemma 12.** *For $d \geq 0$, wide-$\mathsf{TC}^d$ circuit evaluation is hard for $\mathsf{L}$-uniform $\mathsf{TC}^d$ under $\mathsf{L}$-reductions.*

*Proof.* As in the proof of Lemma 11, we are given $L$ such that there exists a log-space Turing machine $T_L$ that constructs $\{C_n\}_{n=0}^{\infty}$ recognizing $L$, where the depth of each $C_n$ is at most $c \log^d n$ for some $c$ and large enough $n$. We can create a modified log-space Turing machine $T_L'$ that builds $\{C_n'\}_{n=0}^{\infty}$ that still recognizes $L$ in depth $c \log^d n$ but is padded, if needed, with dummy gates so that it has size at least $n^c$: we do this by keeping a counter for the number of gates and wires produced and outputting dummy ones until $n^c$ is exceeded. We then follow the rest of the proof of Lemma 11 with $T_L'$ instead of $T_L$. $\qquad\square$

## D.2 Solving Wide-$\mathsf{TC}^d$ Circuit Evaluation with Transformers

**Lemma 13.** *There is a mixed-masked looped transformer $T$ that, on input $(x, \langle C \rangle)$ where $x \in \{0, 1\}^*$ and $\langle C \rangle$ is the serialization of a depth $\ell$ circuit with $|x|$ inputs, computes $C(x)$ when unrolled $\ell$ times.*

*Proof.* We adapt the proof of Merrill and Sabharwal (2023a, Theorem 3), which sketches how log-depth (unlooped) transformers can implement the $\mathsf{TC}^0$ circuit evaluation problem. We construct a looped transformer that will "attempt" to evaluate every gate: if its arguments have already been computed, the gate will return $\{0, 1\}$, and, if not, it will return undefined ($\bot$).

Let $i$ be a token index. We say the token $w_i$ is a gate token if it is X, AND, OR, NOT, MAJ. We will use gate token $i$ to store the value $v_i \in \{0, 1, \bot\}$ for the gate it represents as a one-hot encoded vector. In the base case (embedding layer), we initialize $v_i = \bot$ for every gate token. Using a looped block of 2 layers, we will proceed in a way that propagates the computation of $v_i$ at later layers in terms of previously computed values.

X Gates. In the setup layers at an X token $i$, we use a causally masked uniform attention head with value $\mathbb{1}[w_j = \text{X}]$. Thus, this head computes $r_i/i$, where $r_i$ number of X gates before and including $i$. We compute $\phi(r_i/i, 1/i) = \phi(r_i)$ and store it in the residual stream.

In the looped layers, we define an attention head with query $\phi(r_i)$, key $\phi(j)$, and value $w_j$. This head thus retrieves input token $w_{r_j}$. We update the gate value to $v_i \leftarrow w_{r_j}$ (viewing both as vectors in the same space).

Other Gates. In the setup layers, each argument token $i$ attends with causally masked uniform attention with value $\mathbb{1}[w_j = \&]$ to compute $a_i$, the number of arguments to its left (including it). Each & token attends with query $\phi(a_i)$, key $\phi(a_j)$, and value $\mathbb{1}[w_j = \&]$, which returns $1/(1 + z_i)$, where $z_i$ is the number of 1's following & token $i$. We compute and store $\phi(z_i + 1)$ in the residual stream. We compute $g_i$ similarly to $a_i$, the number of gate tokens to left of token $i$ (also inclusive).

In the first looped layer, each & token $i$ attends with query $\phi(z_i + 1)$, key $\phi(g_j)$, and value $v_j$. Thus, the argument token $i$ retrieves $v_{z_i+1}$, the value at gate $z_i + 1$. In the second looped layer, gate token $i$ attends with query $\phi(g_i)$, key $\phi(g_j)$, and value $v_{z_j+1}$. This, it returns the vector $\frac{1}{|A_i|} \sum_{j \in A_i} v_j$, where $A_i$ is the set of gates that are arguments of gate $i$. From this vector, we can apply projections to recover $T$, the fraction of $j \in A_j$ with $v_j = 1$, as well as $U$, the fraction of $j \in A_i$ with $v_j = \bot$. If $U > 0$, we set $v_i \leftarrow \bot$. Otherwise, we set $v_i$ by thresholding $T$ against some threshold $k$ based on the gate type ($T \geq 1$ for AND, $T > 0$ for OR, and $T \geq 1/2$ for MAJ). In effect, this sets $v_i \leftarrow G(\{v_j\}_{j \in A_j})$, where $G$ is the gate type.

Thus, the looped layers either keep $v_i$ as $\bot$ or correctly update it to its true value. Furthermore, the number of looping steps until $v_i$ is updated is exactly the depth of node $i$. Thus, a circuit $C$ of depth $\ell$ can be fully evaluated by looping the depth-2 block $\ell$ times. $\qquad \square$

**Corollary 13.1.** *For $d \geq 0$, wide-$\mathsf{TC}^d$ circuit evaluation is in $\mathsf{mAHAT}_0^d$, and hence in $\mathsf{AHAT}_1^d$.*

*Proof.* We are given input $(x, \langle C_n \rangle)$, where $C_n$ comes from some wide-$\mathsf{TC}^d$ circuit family $\{C_n\}_{n=0}^{\infty}$ with depth at most $c \log^d n$ and size at least $n^c$, where $c$ is a constant specific to the circuit family. If we unroll the transformer from Lemma 13 to depth $c \log^d n$, we can solve wide-$\mathsf{TC}^d$ circuit evaluation problem for large enough $n$ (w.l.o.g. we can solve small-$n$ examples via table lookup).

We next justify that a mixed-masked $\mathsf{mAHAT}^d$ transformer will unroll at least $c \log^d n$ times for large enough $n$. This transformer will unroll exactly $\log^d N$ times, where $N = n + |\langle C_n \rangle|$ is the total input length for a circuit evaluation instance. Since the size of $C_n$ is at least $n^c$, we have that $N \geq n^c$. Thus, our $\mathsf{mAHAT}_0^d$ transformer unrolls the following number of times:

$$\log^d N \geq \log^d n^c = c^d \log^d n \geq c \log^d n.$$

Thus, our transformer will unroll a sufficient number of times to solve the wide-$\mathsf{TC}^d$ circuit evaluation problem. It follows that this problem is in $\mathsf{mAHAT}_0^d$.

Finally, we conclude via Proposition 1 that this mixed-masked transformer can be converted to a corresponding causally masked padded transformer, placing the problem also in $\mathsf{AHAT}_1^d$. $\qquad \square$

### D.3 Solving Wide-$\mathsf{TC}^d$ Circuit Evaluation with $\mathsf{FO}$-Uniform Circuits

It is immediate that the looped transformer in Corollary 13.1 can be simulated in $\mathsf{L}$-uniform $\mathsf{TC}^d$. Moreover, since Lemma 6 shows looped padded transformers can be simulated by $\mathsf{FO}$-uniform polylog-depth threshold circuit families, we obtain:

**Lemma 14.** *For $d \geq 0$, wide-$\mathsf{TC}^d$ circuit evaluation is in $\mathsf{FO}$-uniform $\mathsf{TC}^d$.*

Combined with Lemma 12, this implies the following completeness results for wide-$\mathsf{TC}^d$ circuit evaluation:

**Corollary 14.1.** *For $d \geq 0$, wide-$\mathsf{TC}^d$ circuit evaluation is complete for $\mathsf{L}$-uniform $\mathsf{TC}^d$ under $\mathsf{L}$ reductions.*

**Corollary 14.2.** *For $d \geq 0$, wide-$\mathsf{TC}^d$ circuit evaluation is complete for $\mathsf{FO}$-uniform $\mathsf{TC}^d$ under $\mathsf{L}$ reductions.*

## E  Uniformity Collapse for Circuit Classes

In general, for classes A, B such that $\mathsf{A} \subseteq \mathsf{B}$, B-uniformity often leads to larger classes of languages than A-uniformity, as we have more resources to construct a circuit. However, as the following lemma shows, this is not always the case:

**Proposition 6** (Uniformity Collapse). *Consider classes A, B of functions and a class XC of polynomial size circuit families such that:*

1. *$\mathsf{A} \subseteq \mathsf{B} \subseteq \mathsf{A}$-uniform XC;*

2. *XC circuit evaluation is in A-uniform XC;*

*3. A-uniform* XC *circuit families are closed under fixed compositions (cf. Appendix B.1).*

*Then* B-*uniformity does not strengthen* A-*uniformity, i.e.,* A-*uniform* XC = B-*uniform* XC.

*Proof.* Since A $\subseteq$ B, we trivially have A-uniform XC $\subseteq$ B-uniform XC. The rest of the proof will focus on the other direction, showing that any language $L \in$ B-uniform XC is also in A-uniform XC.

By the definition of B-uniform XC, there exists a function $f \in$ B that constructs an XC circuit family $\{C_n^f\}_{n \geq 0}$ that recognizes $L$. Specifically, $f(1^n) = C_n^f$ where $C_n^f$ is an XC circuit that checks membership in $L$ over all strings $w$ of size $n$: for $w \in \{0,1\}^n$, $C_n^f(w) = 1$ iff $w \in L$. The key insight is the following: while $\{C_n^f\}_{n \geq 0}$ is not necessarily an A-uniform XC circuit family, we can *build* $C_n^f$ on demand using a different, A-uniform XC circuit family $\{C_n^g\}_{n \geq 0}$, by leveraging the first precondition of the lemma, specifically that $B \subseteq$ A-uniform XC. We will then leverage the second precondition to construct another A-uniform XC circuit family $\{C_{n,m}^h\}_{n,m \geq 0}$ that *evaluates* the built circuit $C_n^f$ on the input string $w$. We will then use the third precondition to compose these two circuit families in order to obtain the final A-uniform XC circuit family $\{C_n^L\}_{n \geq 0}$ that recognizes $L$.

**Building $C_n^f$:** Since $f \in$ B, by the first precondition of the lemma, $f$ is also in A-uniform XC. We can thus simulate $f$ using an A-uniform XC circuit family. More concretely, there exists a function $g \in$ A that, for any $n \geq 0$, maps input $1^n$ to an XC circuit $C_n^g$ that does what $f$ does, i.e., $C_n^g$ on input $1^n$ constructs the circuit $C_n^f$. Thus, we have $g(1^n) = C_n^g$ and $C_n^g(1^n) = C_n^f$.

**Evaluating $C_n^f$ on $w$:** By the second precondition of the proposition, there exists a function $h \in$ A that, for any $n, m \geq 0$, maps input $(1^n, 1^m)$ to an XC circuit $C_{n,m}^h$ that evaluates any XC circuit $C_n$ of size $m$ over any input string $w$ of size $n$, i.e., $h(1^n, 1^m) = C_{n,m}^h$ and $C_{n,m}^h(C_n, w) = C_n(w)$ for $|w| = n$.

**Composing builder and evaluator circuits:** Now we are ready to construct an A-uniform XC circuit family $\{C_n^L\}_{n \geq 0}$ that recognizes $L$. This circuit family, on input $w$ of length $n$, computes the following composition of the circuit builder family $\{C_n^g\}_{n \geq 0}$ and the circuit evaluator family $\{C_{n,m}^h\}_{n,m \geq 0}$:

$$C_n^L(w) = C_{n,m}^h(C_n^g(1^n), w) \tag{1}$$

where $n = |w|$ and $m = |C_n^f|$. We first observe that this composite circuit does, in fact, recognize $L$. That's because for any input string $w$ of length $n$:

$$\begin{aligned} C_n^L(w) &= C_{n,m}^h(C_n^g(1^n), w) \\ &= C_{n,m}^h(C_n^f, w) \\ &= C_n^f(w) \end{aligned}$$

which, by definition, is 1 if and only if $w \in L$. Thus, $C_L$ recognizes $L$.

Lastly, by the third assumption of this proposition, the circuit family $\{C_n^L\}_{n \geq 0}$, being a fixed composition of A-uniform circuit families $\{C_{n,m}^h\}_{n,m \geq 0}$, $\{C_n^g\}_{n \geq 0}$, and the identify function (for constructing the second argument $w$ of $C_{n,m}^h$), is also A-uniform. $\square$

Proposition 6 may appear surprising on the surface level, but it has a quite intuitive interpretation: weaker uniformity provably cannot increase a circuit class that is sufficiently expressive. This has an interesting implication for polylog-depth circuit classes: FO and L-uniform $\mathsf{TC}^d$ collapse for $d \geq 1$:

**Theorem 3** (Uniformity Collapse). *For any $d \geq 1$, the following equivalences hold:*

$$\mathsf{FO}\text{-}uniform\ \mathsf{AC}^d = \mathsf{L}\text{-}uniform\ \mathsf{AC}^d = \mathsf{NL}\text{-}uniform\ \mathsf{AC}^d$$

$$\mathsf{FO}\text{-}uniform\ \mathsf{TC}^d = \mathsf{L}\text{-}uniform\ \mathsf{TC}^d = \mathsf{NL}\text{-}uniform\ \mathsf{TC}^d.$$

*Proof.* We state the proof for $\mathsf{TC}^d$, but everything generalizes for $\mathsf{AC}^d$ as well. By construction, the following containments hold:

$$\mathsf{FO}\text{-uniform}\ \mathsf{TC}^d \subseteq \mathsf{L}\text{-uniform}\ \mathsf{TC}^d \subseteq \mathsf{NL}\text{-uniform}\ \mathsf{TC}^d.$$

We will show that NL-uniform $\mathsf{TC}^d \subseteq$ FO-uniform $\mathsf{TC}^d$ using Proposition 6, with A = FO, B = NL, and XC = wide-$\mathsf{TC}^d$.[5] With A = FO, the third precondition of Proposition 6 follows from Proposition 3. We next argue that its first two preconditions are also met:

1. Since $d \geq 1$ and NL is known to be in FO-uniform $\mathsf{AC}^1$, which is contained in FO-uniform $\mathsf{TC}^1$, we have NL $\subseteq$ FO-uniform $\mathsf{TC}^d$ = FO-uniform wide-$\mathsf{TC}^d$, where the final equality comes from Proposition 5. Thus, the first precondition is satisfied.

2. From Lemma 14, we also have that wide-$\mathsf{TC}^d$ evaluation is in FO-uniform $\mathsf{TC}^d$, which by Proposition 5 equals FO-uniform wide-$\mathsf{TC}^d$. Thus, the second precondition is also satisfied.

We conclude by Proposition 6 that

$$\mathsf{NL}\text{-uniform }\mathsf{TC}^d \subseteq \mathsf{FO}\text{-uniform wide-}\mathsf{TC}^d = \mathsf{FO}\text{-uniform }\mathsf{TC}^d.$$

Thus, for $d \geq 1$, FO-uniform, L-uniform, and NL-uniform $\mathsf{TC}^d$ collapse to the same class of languages. $\qquad\square$

Notably, the proof of Theorem 3 does not go through in the case of $\mathsf{TC}^0$ since it is not known (and not believed) that L and NL are in $\mathsf{TC}^0$.

---

[5]This can be proven analogously for $\mathsf{AC}^d$ and wide-$\mathsf{AC}^d$.

