# OpenReview forum: "Exact Expressive Power of Transformers with Padding"
_NeurIPS.cc/2025/Conference — NeurIPS 2025 poster_

### Official Review · Reviewer_KMSi · 2025-06-26

**Clarity:** 2
**Significance:** 2
**Originality:** 3
**Rating:** 4
**Confidence:** 3

**Summary:**

It was observed previously that average-hard attention transformers with logarithmically many bits can only recognize language from TC^0 (which is almost obvious, they have constant number of layers and arithmetic operations that they employ are doable within TC^0). Moreover, if one uses some simple explicit positional encoding, one can show that such transformers belong to the dlogtime-uniform TC^0.

Does this inclusion hold in the other direction? Intuitively, there is a big obstacle -- transformers have linear size while TC^0 circuits can have arbitrary polynomial size. One of the results of the paper is -- if we are allowed to add polynomially many ``blank'' tokens initially, we obtain a transformer model, exactly equivalent to dlogtime-uniform TC^0.

The paper goes beyond that and asks -- what about NC, the class of polylog-depth circuits? Just adding polynomially many blank tokens does not seem enough since we still have constant number of layers. The paper suggests to allow a fixed constant block of layers to be repeated polylogarithmically many times. They claim that with this, transformers converge precisely to L-uniform NC.

**Questions:**

How precisely you define TC^d? If as O(log^d)-depth threshold circuit, then definition 11 in the supplementary material does not make sense to me, because any circuit belongs to some sequence of circuits from TC^d (by making  constant in O(log^d) big enough).

I guess you should for any constant C, d, define the C,d-threshold circuit evaluation problem, where the input circuit has depth at most Clog^d n precisely.

**Ethical Concerns:**

["NO or VERY MINOR ethics concerns only"]

**Limitations:**

yes

**Paper Formatting Concerns:**

no, but maybe you could simplify the exposition if instead of stating super-precise equivalence like theorem 3 you have stated that L-uniform NC is equivalent to transformers with polylog-looping.  I am not sure that this exact equivalence in the power of log is that interesting on its own.

**Quality:**

3

**Strengths And Weaknesses:**

Characterizing dlogtime-uniform TC^0 via polynomially padded transformers seems valid. This is a solid result but maybe not very surprising and not very exciting technically.

As for the L-uniform NC result, I think there are some technical inaccuracies, but I believe the result should held (at least maybe without precisely equivalence of the hierarchies, just that every language in L-uniform NC can be recognized by a polylog-looped transformer). One non-trivial idea is required here. A naive approach to simulate a polylog-depth circuit by a transformer is to compute each layer of a circuit in a transformer layer. However, in this way we might require to have polylog-many different layers in a transformer, and we just want a loop of a block of a fixed constant number of layers. Instead, the paper constructs a transformer that in a loop, tries to evaluate all gates of a circuit simultaneously, and the gate can be evaluated when all gates that are fed to it are already evaluated. This idea appears in Merrill and Sabharwal 2023a.

Some minor remarks:
Definition 1, h is the number of attention heads, and k is their index?

Line 126, in the positional encoding here, n is the initial length or padded one?

Line 176, maybe clarify that ``unique hard-attention transformers''

Names of Sections 4 and 5 are misleading because you forget to mention uniformity conditions there.

---

> ### Author Rebuttal · Authors · 2025-07-30
>
> > How precisely you define $TC^d$? If as $O(\log^d)$-depth threshold circuit, then definition 11 in the supplementary material does not make sense to me, because any circuit belongs to some sequence of circuits from $TC^d$ (by making constant in $O(\log^d)$ big enough). I guess you should for any constant $C, d$, define the $C,d$-threshold circuit evaluation problem, where the input circuit has depth at most $C \log^d n$ precisely.
>
> Indeed, you’re right that $TC^d$ circuit evaluation is the same as $TC$ circuit evaluation because the constant can be folded in. This is why we have taken the care to define wide-$TC^d$ class of circuits and the wide-$TC^d$ circuit evaluation problem in the appendix.
>
> wide-$TC^d$ is defined as the class of $TC$ circuit families where each family has a fixed $c$ such that depth is at most $c \log n$ and, crucially, size is **at least** $n^c$. With this additional constraint, wide-$TC^d$ circuits express the same languages as $TC^d$, but the evaluation problem can be solved in $TC^d$ (unlike evaluation of $TC^d$, for the reasons you noted). To see why this constraint helps, note that if we increase $c$ to make a wide-$TC^d$ circuit deep, the length of the circuit serialization must increase, which effectively makes the circuit shallow w.r.t. the total input length for the circuit evaluation problem, upper bounding its effective depth by $c^* \log^d n$ for some **fixed** constant $c*$ chosen a priori. This circuit then can be evaluated by a $(c^*  \log^d n)$-depth circuit family (for a fixed $c^*$), which we build for the wide-$TC^d$ circuit evaluation problem.
>
> Thank you for raising this important point. We will clarify this and also make the wide-$TC^d$ a numbered definition (after Definition 11) as it’s an important concept that addresses the issue you mentioned.
>
> > Some minor remarks: Definition 1, $h$ is the number of attention heads, and $k$ is their index?
>
> Yes, we will note this.
>
> > Line 176, maybe clarify that ``unique hard-attention transformers''
>
> Will do.
>
> > Names of Sections 4 and 5 are misleading because you forget to mention uniformity conditions there.
>
> You are right. We omitted the uniformity qualifier to prevent the section titles from becoming too long, but we agree it would be better to include it and will update in revisions.
>
> > Line 126, in the positional encoding here, $n$ is the initial length or padded one?
>
> In discussing this question, we realized that $i/n$ can actually be simplified to $1/i$ embeddings, and all constructions will go through. This works because $1/n$ (where $n$ is the initial input length) can be computed first, and then $\phi(i/n)$ can be computed as $\phi(1/i, 1/n)$.
> This simplifies the conversion from unmasked to masked transformers because $1/i$ encodings can be directly simulated by a uniform causally masked head.  We will update the setup in the paper to use the simpler $1/i$ embeddings and clarify these details. Thanks for your question, which led us to think about this!

---

### Official Review · Reviewer_KzvS · 2025-07-01

**Clarity:** 3
**Significance:** 3
**Originality:** 3
**Rating:** 5
**Confidence:** 3

**Summary:**

This work investigates the impact of padding tokens on the expressive power of Transformers. Focusing on averaging-based hard attention and masked pre-norm log-precise Transformers, the authors demonstrate that polynomial padding enables the model to represent the class $\mathsf{TC}^0$. Additionally, they show that combining padding with looping mechanisms further enhances expressivity: $O(\log^d n)$ looping enables recognition of the class $\mathsf{TC}^d$, while polylogarithmic looping extends this capability to the class $\mathsf{NC}$.

**Questions:**

1. Could the authors provide further discussion on how padding and looping compare to CoT in terms of expressive power (e.g., the differences between the results of this work and those of [1])?

2. Could the authors provide any insights or intuitions on the learnability of the proposed constructions?

[1] Merrill, W. and Sabharwal, A., The Expressive Power of Transformers with Chain of Thought. ICLR 2024.

**Ethical Concerns:**

["NO or VERY MINOR ethics concerns only"]

**Final Justification:**

The authors have satisfactorily addressed my main concerns in their rebuttal. While the learnability limitation remains, this aspect falls outside the paper's core contributions. Given the overall quality of the work and the response, I maintain my positive score.

**Quality:**

3

**Strengths And Weaknesses:**

Strengths:
1. The work consider the practical techniques, i.e., padding and looping, from a theoretical perspective, offering valuable insights into why these methods can be effective.
2. It establishes connections between the expressive power of padded Transformers and well-known complexity classes, providing a precise and hierarchical characterization of their representational capacity.

Weaknesses:
1. The analysis does not compare the benefits and limitations of padding and looping with other reasoning techniques, such as CoT.
2. The theoretical results focus solely on expressivity and do not address the learnability of the proposed constructions.

---

> ### Author Rebuttal · Authors · 2025-07-30
>
> Thank you for your review! You raise a good point regarding the* comparison of padding and looping with chain of thought*. In the case of $\log n$ looping vs. CoT, it is now known [1][2]] that transformers with logarithmic CoT steps remain in $TC^0$ whereas log-looped transformers can solve some $NC^1$-complete problems, suggesting an advantage of looping. More generally, the comparison between transformers with $t$ steps of looping vs. $t$ steps of CoT remains open. We will update the paper to discuss this comparison, what is known, as well as the open question.
>
> Regarding *learnability*, we agree this is very interesting, but is beyond the scope of the current work, which we can acknowledge as a limitation.
>
> [1] https://arxiv.org/abs/2503.03961
> [2] https://arxiv.org/abs/2210.10749

---

> > ### Comment · Reviewer_KzvS · 2025-08-06
> >
> > Thank you for your response. I appreciate the clarifications, which satisfactorily address my concerns. I will maintain my positive score.

---

### Official Review · Reviewer_FTuT · 2025-07-03

**Clarity:** 4
**Significance:** 3
**Originality:** 4
**Rating:** 6
**Confidence:** 4

**Summary:**

This paper investigates the expressive power of Transformers when augmented with two parallelizable forms of inference-time compute: padding and looping. The authors provide a precise formal characterization of the class of problems solvable by these models. Their first main result is that fixed-depth, polynomially padded Transformers can recognize exactly the complexity class `TC^0`, a class of highly parallelizable problems. This result resolves an open question, as previous work had only established `TC^0` as an upper bound. The second main result extends this analysis to Transformers that dynamically repeat a block of layers. The paper shows that a Transformer with polynomial padding and `O(log^d n)` looping recognizes exactly the class `L-uniform TC^d`. This establishes a systematic relationship between the depth of looping and the computational power of the model, showing that these parallelizable mechanisms can expand a Transformer's capabilities to encompass the entire class `NC` (the class of efficiently parallelizable problems), a significant increase in power over un-augmented models.

**Questions:**

1.  **On the Practicality of Padding:** The constructions require `O(n^k)` padding to simulate `k` variables, which can become very large. Could you comment on the prospects for reducing this padding requirement? For instance, are there more compressed ways to represent the variable assignments, or could a model potentially learn to use a smaller scratchpad more efficiently? Is the `n^k` dependency a hard theoretical barrier for your proof technique, or do you see some approaches that can tighten this?

2.  **Bridging the Gap to Soft Attention:** Could you elaborate on the key challenges in extending your results from the AHAT model to standard soft-attention Transformers? The core of your proof seems to rely on precise retrieval and counting via hard attention. What kind of new techniques or assumptions would be needed to show that soft attention can approximate these operations with sufficient fidelity, especially as sequence length `n` grows?

3.  **The Role of Uniformity:** The paper proves equivalence with `L-uniform TC^d`. The appendix notes that this implies `L-uniform TC^d = NL-uniform TC^d`. This is an interesting complexity-theoretic side effect of the analysis. Could you elaborate slightly on why your Transformer construction naturally leads to `L` or `NL` uniformity rather than, for example, the stricter `FO`-uniformity seen in the `TC^0` case? Is it because solving the complete problem (e.g., Graph Connectivity) requires the power of `L`?

4.  **Implications for Architecture Design:** Your results provide a compelling theoretical motivation for using padding and looping. Do you have any high-level thoughts on how these findings might inform future Transformer architecture design? For example, would it make sense to design models with dedicated "scratchpad" blocks or explicit looping mechanisms that are more suitable to learning the kinds of algorithms you describe?

**Ethical Concerns:**

["NO or VERY MINOR ethics concerns only"]

**Final Justification:**

Rating stays the same. Oral quality paper in my opinion.

The paper is of top quality, all the comments are addressed, and I keep my rating of Strong Accept.

**Limitations:**

Yes, the authors have adequately addressed the limitations.

**Paper Formatting Concerns:**

None.

**Quality:**

4

**Strengths And Weaknesses:**

**Strengths:**

1.  **Significant Impact:** The paper provides a definitive and exact characterization of the expressive power of padded and looped Transformers. This closes an important gap in the theoretical understanding of Transformers and provides a formal basis for exploring alternatives to sequential reasoning methods like Chain of Thought (CoT).

2.  **Technical Quality and Rigor:** The paper is technically flawless. The proofs are carefully constructed, building from the base case of fixed-depth padded Transformers to the more general case of looped models. The authors clearly define their idealized model (AHAT) and leverage its properties to build a rigorous, step-by-step argument. The connection to string logics (`FO[M^2]`) to circumvent the `BIT` predicate issue is original and demonstrates a deep understanding of the relevant literature in both machine learning and complexity theory.

3.  **Clarity of Presentation:** Despite the technical density of the material, the paper is exceptionally well-written. The introduction clearly motivates the problem and situates the work within the existing literature. The main results are stated cleanly, and the proof sketches provide excellent intuition for the formal constructions. Figure 1 is a nice summary of the paper's contributions and how they relate to known complexity classes.

**Weaknesses:**

1.  **Idealized Model:** The analysis is based on an Averaging-Hard-Attention Transformer (AHAT) with masked pre-norm and `O(log n)` precision. While this is a common and necessary simplification in theoretical work, it creates a gap between the theoretical model and the soft-attention, fixed-precision Transformers used in practice. The authors acknowledge this, but the practical implications of their findings hinge on the assumption that these results will translate, which is not guaranteed.

2.  **Lack of Empirical Validation:** The paper is purely theoretical. While the goal is to establish formal limits of computation, the practical feasibility of the proposed constructions remains an open question. For instance, the polynomial padding required (`n^k` tokens) could be prohibitively large for even moderate `k`. The paper would be even stronger if it included a small-scale empirical demonstration or a more in-depth discussion of how these theoretical ideas might be approximated in practice.

3.  **Learnability:** The paper focuses on expressive power (what is computable in principle), not on learnability (what can be learned from data via gradient descent). The constructions are intricate and algorithmic, and it is unclear if a standard training procedure could ever converge to the specific weight configurations required.

---

> ### Author Rebuttal · Authors · 2025-07-30
>
> Thanks for your review! We are happy to hear that you appreciate the technical quality and clarity of our paper as well as its potential for impact.
>
> We agree that the *practical learnability* of AHAT constructions (with masked pre-norm) is an interesting open question. We will add some discussion either in the main paper or in a limitations section. The major gap between soft attention and AHAT is indeed the ability to implement hard-attention retrieval. For any fixed maximum context length, it is possible to scale the attention head parameters to arbitrarily approximate AHAT attention, but for unbounded context lengths, it seems difficult (and may likely be impossible) to simulate AHAT or hard attention. For this reason, we view AHAT as a mostly mild assumption (suitable for bounded context lenghts) that abstracts away issues with soft attention for simulating hard attention over very long contexts.
>
> Regarding the *practicality of padding*, $n^k$ is indeed a practical barrier if $k$ must be large for some tasks. Small values of $k$ will suffice for some natural tasks tasks: for example, in the 3-SUM task considered by Pfau et al. But, in general, there may be tasks in $TC^0$ that require a large value of $k$, which would lead to a large memory overhead to store a long context window (the same problem would plague transformers with long CoT or chain of thought). More work is needed to assess the tasks for which small values of $k$ would suffice or whether other techniques could be developed to reduce the memory overhead of padding -- we will acknowledge this as a limitation.
>
> Regarding your question about $L$ vs. $FO$ uniformity, we have a brief discussion of this in the appendix after the proof of Lemma 4 (that $\log^d$ depth transformers are in $TC^d$).
>
> Indeed, for $d \geq 1$, it can be shown that $\log^d$-looped transformers would be in $FO$-uniform $TC^d$, similar to how you suggest. The idea is that looping an $FO$-uniform $TC^0$ circuit $O(\log^d n)$ times can be implemented in $FO$-uniform $TC^d$.
>
> This has the interesting consequence that $FO$-uniform $TC^d = L$-uniform $TC^d$ by our *uniformity ceiling lemma* (Proposition 2, Appendix C) , just like the $L$-uniform vs. $NL$-uniform collapse noted in the appendix.  In answering your question, we looked closely at the uniformity ceiling argument (Prop. 2) and we think it also goes through when $XC = wide-TC^d$ for $d \geq 1$, class $B = L$, and class $A = FO$, showing that $L$-uniformity collapses to $FO$-uniformity for deeper $TC$ classes.
>
> Note that this doesn’t go through for $TC^0 (d=0)$ because $L$ is not known to be a subset of $FO$-uniform $TC^0$. Thus, the uniformity ceiling lemma cannot be applied.
>
> Thank you for bringing this refinement to our attention! We will add a brief section highlighting these novel results about uniformity and also the novel complete problems we developed to obtain these uniformity collapse results, since these might be of independent interest.

---

> > ### Comment · Reviewer_FTuT · 2025-08-02
> > **Rating stays the same. Oral quality paper in my opinion.**
> >
> > The paper is of top quality, all the comments are addressed, and I keep my rating of Strong Accept.

---

### Official Review · Reviewer_YvVK · 2025-07-07

**Clarity:** 2
**Significance:** 3
**Originality:** 3
**Rating:** 4
**Confidence:** 3

**Summary:**

This paper precisely characterizes the expressive power of transformers with a polynomial number of _padding_ tokens by providing a matching lower bound on their ability to solve $TC^0$ problems. Additionally, the authors show that $O(\log^d n)$ _looping_ layers enable padded transformers to be precisely characterized by $TC^d$. Using padding effectively enlarges the width of the transformer, while looping effectively increases the depth of the transformer. While the expressive power of padded and looped transformers remains below the expressive power of chain-of-thought (COT) transformers, they are more easily parallelizable and thus offer computational advantages in comparison to COT.

**Questions:**

1) To my current understanding, Proposition 10.3 in Barrington et al. implies that we can express FO[M$^2$] by using formulas from FO[M, $\texttt{bit}$], but not the other way around. Could you clarify what "FO[M$^2$] ... is equivalent to standard FO[M, $\texttt{bit}$]" means in this context?
2) In lines 352-357 in the proof of Lemma 3: Why do we check if $f(w) \in L'$ instead of $L$?
3) In Theorem 1 and Theorem 2, do we assume infinitely many padding tokens (given Definition 6), and thus infinite width?
4) Could you explain what exactly is meant by the phrase "converge to recognizing NC" in the context of your theoretical results?
5) How expensive is padding and looping in practice? How can padding be efficiently parallelized?

Barrington, D. A. M., Immerman, N., & Straubing, H. (1990). On uniformity within NC1. Journal of Computer and System Sciences, 41(3), 274-306.

**Ethical Concerns:**

["NO or VERY MINOR ethics concerns only"]

**Final Justification:**

I did not increase my score since, as also acknowledged by the authors, the paper has some issues with clarity, which makes it difficult for me to judge its overall soundness. However, given the confident review by reviewer FTuT which attests the paper to be "technically flawless", I assume that all the proofs were carefully checked and are correct (except for the discussed mistakes in notation). This, together with the novelty and significance of the theoretical contributions, makes me lean towards acceptance.

**Limitations:**

There is no explicit discussion of limitations.

**Quality:**

2

**Strengths And Weaknesses:**

*Strengths*: The submission provides novel bounds for transformers with padding as well as looping in terms of circuit complexity classes, which helps compare their expressive power to other transformer architectures. Additionally, Lemma 3 introduces the concept of _reductions_ for analyzing the expressive power of transformers, which might be useful on its own to derive further results.

*Weaknesses*: There are several issues with clarity, which makes it generally difficult to assess the soundness of the submission. While the paper is easy to read, it would benefit from more rigorous definitions and careful proofreading. In several places, the presentation appears imprecise. For example, $\phi$ is used inconsistently, sometimes as a formula (without being introduced as such), and elsewhere as a layer norm. Additionally, the use of colloquial language and frequent quotation marks around technical terms contributes to ambiguity. For instance, in line 310, why is 'token' in quotes; similarly, line 247 mentions a "projection," and line 40 refers to "storage space," both again in quotation marks; in the proof of Lemma 3 in line 354, the phrase "very negative" is vague. Overall, the proofs rely heavily on informal language, which hinders accessibility and undermines precision. Without more rigorous definitions and terminology, the arguments are difficult to follow for me. Lastly, the absence of a dedicated related work section makes it more difficult to situate the contributions within the broader context.

Please find more detailed remarks below:

- The variable $\phi$ appears many times in seemingly different contexts:
    * $\phi(i)$ in line 111
    * $\phi$ in Definition 7, 5(b)
    * $\phi(i,j)$ in Definition 8
    * $\phi(b_i - 1)$ in line 209
    * $\phi(i)$ in line 343
- Line 123 in Definition 6: What does universal mean in this context, looped?
- Line 142: "for" instead of "For"
- Line 166: "other" instead of "over"
- First sentence in Definition 7 should be outside of definition environment
- Line 211: What is the $\texttt{bos}$ token?
- Line 216: What does "masked unmasked head" mean?
- The proof of Lemma 2 is difficult to follow:
    * Many terms seemingly refer to the same variable: $v$ is a token (lines 240,245), a variable assignment (line 240), an integer (line 246), and a configuration (line 250).
    * Line 233: Should this be Definition 7 instead of Definition 8?
    * Could you explain what "Each token corresponds to a specific assignment of all the variables, which we denote $v$" means?
    * Lines 261 and 268: Why do we have $[[P]]^{w,v}$ in line 261 and $P^v$ in line 268?
    * Lines 265 and 272: What is $v'$?

---

> ### Author Rebuttal · Authors · 2025-07-30
>
> Thank you for your review!
>
> Thanks for your careful reading and *feedback on clarity*. To respond to some of your points:
>
> * Regarding $\phi$, we will standardize it to only refer to the layer-norm hash. We will change Definition 3 to explicitly mention this sense of $\phi$ (we apologize that it seems to have gotten removed at the last minute) We will change the symbol for formulas in Definitions 5 and 8 to $P$ or $Q$. This will make both aspects of the paper clearer to read.
>
> * In Definition 6, “universal” indeed means “looped”. We will change this. By “bos” we mean a beginning-of-sequence symbol \$ as defined on page 3. We will update the text to make this clear.
>
> * “Masked unmasked” is a typo and should just be “masked”
>
> Regarding Lemma 2, we will take into account your feedback on clarity, and briefly address here your following question:
>
> > Could you explain what "Each token corresponds to a specific assignment of all the variables, which we denote $v$" means?
>
> If a formula has $k$ variables ranging from $1$ to $n$, there are $n^k$ possible assignments to that vector of variables. We can think about a single assignment $v$ as a tuple in $[n]^k$, or we can think about it as a number between $1$ and $n^k$. What we mean is that we identify each variable assignment with a specific number. Then, token $v$ will represent variable assignment $v$.
>
> We will also incorporate all the other detailed feedback you mentioned regarding clarity and informal language - thanks for reading carefully! On the use of informal terms, we did that in certain places to make the argument more accessible to broader audience. However, we see that it can result in lack of precision and we will make sure to make the language more precise -- critically in all proofs.
>
> > To my current understanding, Proposition 10.3 in Barrington et al. implies that we can express $FO[M^2]$ by using formulas from $FO[M,Bit]$, but not the other way around…
>
> Your interpretation of Proposition 10.3 is correct, but what we use here is Theorem 10.2, which proves the other direction that $FO[M^2]$ can capture all of $TC^0$. We will update our reference to the Barrington et al. paper to include the relevant theorem.
>
> > In lines 352-357 in the proof of Lemma 3: Why do we check if $f(w) \in L’$ instead of $L$?
>
> Indeed, have a minor issue with the notation in this paragraph where we flipped L and L’. We will correct it to say that we are checking whether $f(w) \in L$, which is equivalent to checking $w \in L’$. Thanks for finding this!
>
> > In Theorem 1 and Theorem 2, do we assume infinitely many padding tokens (given Definition 6), and thus infinite width?
>
> The number of padding tokens needed in the simulations in Theorem 1, 2, and 3 is naturally a function of the complexity of the underlying $TC^0$, $NL$, and $TC^d$ circuits / algorithms. It’s not infinite, but rather *unbounded* a priori. E.g., if a $TC^0$ circuit being simulated is equivalent to a $FO[M^2]$ formula with $k$ distinct variables, then we need $n^k$ padding tokens (in Lemma 2 and Theorem 1). Since one can define arbitrarily large circuits, their transformer simulation also needs to support correspondingly more (but still polynomially many) padding tokens, i.e., a larger exponent $k$ in $n^k$. Crucially, for every problem, there is a fixed $k$ that works. The subscript $$ in our notation $AHAT^d_*$, representing the union over all $k$, captures this. We will clarify this, in particular that for simulating any *fixed* circuit family, we need only a fixed polynomial degree number of padding tokens.
>
> > Could you explain what exactly is meant by the phrase "converge to recognizing NC" in the context of your theoretical results?
>
> $NC$ can be defined as $\bigcup_{k = 0}^\infty TC^k$. With $O(\log^k)$ depth, we showed padded transformers can recognize $TC^k$. So, as $k \to \infty$, $O(\log^k)$ depth (i.e., polylog-depth) transformers can recognize any problem in $NC$. In other words, for any problem in $NC$, there is *some* choice of $k$ such that a $\log^k$ depth padded transformer will recognize it. We will clarify the discussion around this to make it more precise.
>
> > How expensive is padding and looping in practice? How can padding be efficiently parallelized?
>
> Padding is efficiently parallelizable in practice: in can be thought of as just growing the input with blank symbols, and then making a single call for transformer inference (as opposed to CoT, which requires sequential inference calls to the transformer).
>
> The practical bottleneck it would run into is the length of the context window, as running a transformer with a long context will run into memory limitations. Some problems like 3SUM considered by Pfau et al. are solvable with a small padding degree $k$, but if a problem requires a larger value of $k$, this could definitely become impractical (the same argument holds for CoT, in addition to the sequentiality of CoT). We will clarify this potential limitation in the text.

---

> > ### Comment · Reviewer_YvVK · 2025-08-07
> >
> > Thank you for the clarifications! I will keep my current score, but lean towards acceptance with the understanding that the notational mistakes and imprecisions will be fixed for the camera-ready version.

---

### Decision · Program_Chairs · 2025-09-17

**Decision:**

Accept (poster)

**Comment:**

**Scientific Claims and Findings**
This paper provides a precise characterization of the expressive power of Transformers augmented with padding and looping, two parallelizable inference-time computation mechanisms. The authors' central claim is that fixed-depth, hard-attention Transformers with a polynomial number of padding tokens can recognize exactly the complexity class $TC^0$. The paper further demonstrates that by incorporating $O(\log^d n)$ looping (dynamically repeating a block of layers), the expressive power is precisely extended to the class $TC^d$. Consequently, with polylogarithmic looping, these models can recognize the entire class $NC$, which represents the class of efficiently parallelizable problems.

**Strengths**
This paper definitively resolves an open question by providing an exact, rather than an upper-bound, characterization of padded Transformers' expressive power, which significantly deepens our understanding of expressiveness of transformer architecture. The reviewers also praise the clarity of the writing.

**Weaknesses**
1. The main weakness of the submission is that the finite precision number system and thus the main function class considered in the paper $AHAT^d_k$ are not defined in the paper. This does not directly hurt the validity of the main result of the paper, that the looped transformers with polynomially many padding can simulate $TC^k$ circuits. However, without a formal and rigorous definition of the function class $AHAT^d_k$, the other direction of equivalence, that circuits can simulate transformers, is ambiguous. The main issue comes from the imprecision about O(log n) precision number system.

     First, it is clear that without any approximation, it is hard to simulate the embeddings of the transformer using TC circuits with perfect precision, because the \sqrt operation in layernorm ill-defined as rounding is necessary for \sqrt. The average hard attention also allows the transformer to take average of poly(n) numbers, which drastically increases the precision for exact simulation. So the natural fix here seems to define the forward pass of transformer as **first compute exactly and then round to finite precision** for every basic operation such as square root, multiplication and addition over poly(n) numbers. Though I think in principle a rigorous and relatively simple fix based on "exact computation and then round" exists, there are a few subtle issues that remains and I would like the authors to be aware:

     (1). details of the number system and rounding: how do you represent a rational number? is it in the form of p/q, where p,q are integers bounded by poly(n) (so O(log n) bits), or p/2^x, where p,x is integers with absolute values bounded by poly(n) (the latter one is more like floating point numbers)? do you always around to the closest representable number, or your round to any number that is sufficiently close? (but then to make the output a well-defined deterministic function class, you have to show that the potential non-determinism does not output the output token nondeterministic, in some sense, a margin-based analysis is required)

    (2). the (potential) increased complexity of the proof of the main direction -- transformer simulates circuits, due to the existence of rounding. Here a careful control of rounding error and its propagation is needed, especially in the polylog(n) depth case.

    (3). Depending on the representation system the authors choose, the actual needed precision of forward pass could be much higher than O(log n). For example, suppose rational numbers are stored in the form of p/q, where p,q are integers bounded by poly(n) (so O(log n) bits), then the sum of M such numbers could require $\Omega(M\cdot \log n)$ bits to represent. When M =poly(n), the proposed mathematical model of transformers $AHAT^d_k$ might need to work with a much higher precision than O(\log n), which is $\Omega(poly(n))$.

2. Other weaknesses identified by reviewers are including the reliance on an **idealized Transformer model** (Averaging-Hard-Attention Transformer, AHAT) as opposed to the more practical version (finite precision, softmax attantion) and the **lack of empirical validation or discussion of learnability**. The theoretical constructions, which require specific weight configurations and potentially a large number ($n^k$) of padding tokens, may not be achievable through standard gradient-based training or practical for real-world applications.

**Reasons for Recommendation**
Overall, the work makes a significant contribution to the theoretical understanding of Transformers. By precisely linking the architecture of padded and looped Transformers to well-established complexity classes ($TC^0$, $TC^d$, and $NC$), it provides a theoretical foundation for exploring parallelizable alternatives to sequential reasoning methods like Chain of Thought. The paper's technical depth and the resolution of an open problem make it stand out as a significant advance in the field, if the above main weakness can be addressed.

The final recommendation for this paper is Accept(poster), conditioned on that the authors will give a formal and rigorous definition for the function class $AHAT^d_k$ and rigorous proofs for the both directions of the main results. This is the conclusion that SAC and AC have reached after a long discussion and is based on our impression that a rigorous and relatively simple fix based on "exact computation and then round" exists. It is of course up to the authors to choose their way to fix the current issues, and we expect the authors to withdraw the paper if they cannot make it precise and rigorous. In particular, I recommend that the authors write as explicit about the number system and forward pass of transformers as possible, and try to avoid use big O notations in the key definitions about the function class.